# Hydroclimatic Variability and Predictability: A Survey of Recent Research

Randal D. Koster[1], Alan K. Betts[2], Paul A. Dirmeyer[3], Marc Bierkens[4], Katrina E. Bennett[5], Stephen J. Déry[6], Jason P. Evans[7], Rong Fu[8], Felipe Hernandez[9], L. Ruby Leung[10], Xu Liang[9], Muhammad Masood[11], Hubert Savenije[12], Guiling Wang[13], and Xing Yuan[14]

[1]Global Modeling and Assimilation Office, NASA/GSFC, Greenbelt, MD, USA
[2]Atmospheric Research, Pittsford, VT, USA
[3]Center for Ocean–Land–Atmosphere Studies, George Mason University, Fairfax, Virginia, USA
[4]Department of Physical Geography, Utrecht University, The Netherlands
[5]Earth and Environmental Sciences, Los Alamos National Lab, Los Alamos, NM, USA
[6]Environmental Science and Engineering Program, University of Northern British Columbia, Prince George, British Columbia, Canada
[7] Climate Change Research Centre and ARC Centre of Excellence for Climate System Science, UNSW, Sydney, New South Wales, Australia
[8]Department of Atmospheric and Oceanic Sciences University of California, Los Angeles, CA, USA
[9]Department of Civil and Environmental Engineering, University of Pittsburgh, PA, USA
[10]Atmospheric Sciences and Global Change Division, Pacific Northwest National Laboratory, P.O. Box 999, Richland, WA, USA
[11]Bangladesh Water Development Board (BWDB), Design Circle – 1, Dhaka, Bangladesh
[12]Water Resources Section, Faculty of Civil Engineering and Geosciences, Delft University of Technology, Stevinweg 1, 2628 CN Delft, The Netherlands
[13]Department of Civil & Environmental Engineering and Center for Environmental Science and Engineering, University of Connecticut, Storrs, CT, USA
[14]CAS Key Laboratory of Regional Climate-Environment for Temperate East Asia (RCE-TEA), Institute of Atmospheric Physics, Chinese Academy of Sciences, Beijing 100029, China.

*Correspondence to*: Randal D. Koster (randal.d.koster@nasa.gov)

**Abstract.** Recent research in large-scale hydroclimatic variability is surveyed, focusing on five topics: (i) variability in general, (ii) droughts, (iii) floods, (iv) land-atmosphere coupling, and (v) hydroclimatic prediction. Each surveyed topic is supplemented by illustrative examples of recent research, as presented at a 2016 symposium honoring the career of Professor Eric Wood. Taken together, the recent literature and the illustrative examples clearly show that current research into hydroclimatic variability is strong, vibrant, and multifaceted.

.

## 1 Introduction

Drought has been linked to the collapse of several ancient societies, including Mesopotamia's Akkadian empire (Cullen et al. 2000), late Bronze-Age cultures in the Eastern Mediterranean (Kaniewski et al 2013), and the Mayan (Haug et al. 2003), Mochica, Tiwanaku and Anasazi civilizations (deMenocal 2001). Flooding may have contributed to the decline of the Cahokia

settlement in the Mississippi River floodplain near modern-day St. Louis about a thousand years ago (Benson et al. 2007; Munoz et al. 2015). While these particular societal impacts of hydrological variability are rather extreme, more moderate and common impacts of the variability are still profound. Droughts continue to generate tremendous economic losses across the globe through their impacts on crop productivity and water supply. Flooding causes extensive damage worldwide; the flooding

of the Mississippi River in 1993, for example, caused over 15 billion dollars of damage (NOAA, 1994). Even minor hydrological variations are becoming ever more relevant in the face of increasing populations across the globe and concomitant reductions in water quality.

Humans, attuned to such vulnerability, have been quantifying hydrological variability and its impacts on society for millennia. Dooge (1988) notes that thousands of years ago, specific and quantified stages of the Nile were tied to hunger (drought) at the

low end and to disaster (flooding) at the high end. Leonardo da Vinci documented floods on the Arno River, driving him to formulate some of the first scientifically-based theories of hydrological variability (Pfister et al. 2005). Humans have long struggled, in fact, to control hydrological variations and thereby mitigate their negative impacts. Over the centuries, reservoirs have been built specifically to provide water to society during dry periods and to serve as a buffer against flooding during pluvial periods, and reservoir operation algorithms have evolved to optimize their effectiveness for both roles. More recently,

techniques have been devised for quantified predictions of hydrological variations. Seasonal streamflow predictions, for example, are tied to snowpack, soil moisture, and climatic state (e.g., Maurer and Lettenmaier, 2003). Precipitation forecasts have become an essential product of operational seasonal forecasting systems (NRC, 2010). Such predictions, if accurate, can inform water management and can help society prepare for some of the more costly and dangerous manifestations of hydrological variation.

Analyses of large-scale hydrological variations and our ability to predict them underlie much of the science of hydroclimatology, the study of the hydrological cycle in the context of the global climate system. While much valuable work on hydrology and hydrological prediction still occurs at catchment and smaller scales (e.g., Abrahart et al. 2012, Wang et al. 2015), the need for a global-scale perspective – one not limited by either political or catchment boundaries – has long been recognized (e.g., Eagleson 1986, Dirmeyer et al. 2009), and this perspective continues to grow in importance. Many important

hydrological problems must be addressed at the large basin scale, a scale that transcends political boundaries and is not amenable to techniques designed for traditional small-scale catchments. Consider also that if meteorological drought (i.e., a rainfall deficit) is ever to be predicted, it would be through consideration of the connections, via the atmospheric circulation, between the local rainfall and the large-scale spatial patterns of ocean and land conditions. Another topic requiring a global-scale perspective is anthropogenic climate change, which has the potential to produce significant changes in the large-scale

hydrological cycle and thus in local hydrological variability. Such impacts raise serious, pressing questions about the sustainability of society's water resources and further underline the need to solidify our understanding of hydrological variations and what controls them (Jiménez Cisneros et al. 2014).

Global-scale modeling systems are critical tools for large-scale hydroclimatic studies. Gridded models of land surface processes driven with meteorological forcing derived from decades of observational data allow the characterization of hydrological variability across extensive time and space scales. When such gridded land models are combined with numerical models of atmospheric and oceanic processes, simulations of the global climate system itself are possible. Such climate simulations can have tremendous value; they can reveal how the different facets of the global hydrological cycle connect to each other, and understanding such connections is essential to our hopes for predicting drought and other manifestations of large scale hydroclimatic variability. Critical limitations to such studies are deficiencies in the models' abilities to capture teleconnections existing in nature (the effect of variations in one part of the system on remote variations in another, such as the impact of the El Niño cycle on continental precipitation) and, as a result, the improvement of these models has long been a high priority research topic. As with hydroclimatic science itself, the complexity and richness of large-scale models has been growing steadily with time.

A large cross-section of hydrologists and hydroclimatologists met in June 2016 at a symposium in Princeton, New Jersey, USA to honor the career of Professor Eric Wood, and the broad range of topics covered in the symposium touch on many of these aspects of large-scale hydrological variability. Given these contributions, and given the ever-evolving state of this important subject, the gathering was seen as an opportunity to survey recent, relevant state-of-the-art hydroclimatic research. We provide such a survey in the present paper, recognizing the fact that hydroclimatological research is but a subset of the much broader range of research underlying the science of hydrology. Here we specifically emphasize research of a large-scale nature; we do not pretend to cover the extensive work being performed, for example, at or below the catchment scale.

In this paper, for each of a number of subtopics relevant to large-scale hydrological variability (namely, general variability and trends, droughts, floods, land-atmosphere interaction, and hydrological prediction), we briefly summarize some findings in the recent literature, going back to about 2010. The survey, while not exhaustive, should serve to provide interested readers with multiple starting points for further study. For each subtopic, we also provide some state-of-the-science findings that were presented at the symposium. Each of these findings is presented in the form of a self-contained, stand-alone figure and caption; together, the figures illustrate the many facets of hydrological variability and the variety of approaches used to investigate it.

## 2 Recent Advances in Hydrological Variability and Predictability

### 2.1 General Studies on Variability and Trends

#### 2.1.1 Recent Literature

The last several years of research into the characterization of Earth's hydroclimatic variability reflect, to some extent, two key facets of the problem: (i) the continually growing availability of powerful computational tools (along with more extensive

observational records and improved analysis techniques) for examining this variability, and (ii) the potential for changes in this variability with changes in the global climate. Amongst the most important modern computational tools, at least for continental- or global-scale hydroclimatic analyses, is atmospheric reanalysis: a mathematically optimal blending of modeling and observations that produces complete fields in space and time of important hydrological variables (e.g., Kanamitsu et al.

2002, Dee et al. 2011, Bosilovich et al 2015, Kobayashi et al. 2015; see also https://reanalyses.org/). Collow et al. (2016), for example, utilize a global reanalysis to characterize the dynamical evolution of meteorological variables during the lifecycle of extreme storms in the Northeast United States, and Maussion et al. (2014) use a regional reanalysis to examine precipitation variability over the Tibetan Plateau, linking it, for example, to certain features of the overlying atmospheric circulation. Of course, reanalyses are far from perfect; Trenberth et al. (2011) indicate disparities between the different reanalyses in their

treatments of large-scale moisture transports and associated hydrological variables such as streamflow.

Another computational tool used heavily in the last decade for continental- or global-scale hydrological analysis is the "land data assimilation system", or LDAS, which is basically a gridded array of land model elements driven with observations-based meteorological forcing, some of which is derived from reanalyses. Explored early on by Dirmeyer et al. (2006), more recent applications of the LDAS approach have benefitted from improved global forcing datasets (e.g., Sheffield et al. 2006, Weedon

et al. 2011) and accordingly provide improved descriptions of large-scale land surface hydrology and its variations (Reichle et al. 2011, Xia et al. 2012, Balsamo et al. 2015). Wood et al. (2011) emphasize the importance to society of developing hyper-resolution (≤1 km resolution) land surface modeling systems at continental to global scales; such resolutions would allow an improved representation of the impacts of spatial heterogeneity in surface properties on large-scale hydrological and atmospheric dynamics.

A climate model in "free-running" mode (i.e., without the assimilation of observational data) is a computational tool with a special role in hydroclimatic analysis, being particularly suitable for sensitivity analyses and for analyses requiring extensive (e.g., multi-century) climate data. Using such a model, for example, Tierney et al. (2013) show a connection between Indian Ocean sea surface temperatures (SSTs) and East African rainfall on multi-decadal timescales through the impact of the SSTs on the Walker circulation. Indeed, the second topic noted above (the idea that hydroclimatic variability is changing with time)

is now largely being addressed through sensitivity studies using such climate models. With climate models, one can artificially modify the concentration of $CO_2$ in the atmosphere, among other climate elements, and quantify the model's long term responses. Dirmeyer et al. (2014a), for example, analyze projected water cycle changes in the Coupled Model Intercomparison Project Phase 5 (CMIP5; a climate evolution experiment involving multiple climate drivers performed by dozens of climate modeling groups) and find that a strongly warmed climate may lead to significant increases in drought and flood risk.

Orlowsky and Seneviratne (2012) point to difficulties in extracting hydrological trends from the CMIP5 results but nevertheless find some robust signals, including $CO_2$-induced increases in drought frequency in regions such as the Mediterranean, South Africa, and Central America.

One of the expectations of a warming climate, supported by such modeling studies (e.g., Held and Soden, 2006; Chou and Lan 2012, Kumar et al. 2013), is that currently dry areas will get drier and wet areas will get wetter. One manifestation of such a trend is the narrowing of the Intertropical Convergence Zones (ITCZ) and the expansion of the drier subtropical area (e.g., Su et al. 2014; Lau and Kim 2015); such a change appears to broadly resemble the observed change in the past several decades
(e.g., Wilcox et al. 2012, Fu 2015), which contributed to the shortening of both North and South American monsoon seasons (Arias et al. 2015). However, Greve et al. (2014), upon examining multiple long-term observational datasets, conclude that the "dry gets drier, wet gets wetter" paradigm is not consistently supported by the historical data, at least over land.

Coumou and Rahmstorf (2012) cite numerous studies documenting recent rainfall and storm extremes that, taken together, suggest that greenhouse warming has affected their frequency. An observation-based analysis of global evapotranspiration
fields indicates a positive trend between 1982 and 1997 that has declined thereafter (Jung et al. 2010). A similar evapotranspiration trend change in regions of North America was attributed to variability of precipitation amount (Parr et al., 2016), while Miralles et al. (2013) point to the El Niño cycle as a major control over global-scale evapotranspiration variability. Milly and Dunne (2016) warn that some estimates in the literature of increased potential evapotranspiration (PET) in a warming climate may be excessive, even those that rely on the well-considered Penman-Monteith equation for estimating PET (Monteith
15  1965).

Trends in streamflow are of critical relevance to water management and have been evaluated recently (largely with historical data) in many areas (see Lorenzo-Lacruz et al. [2012] and references therein). Milly et al. (2008) argue that the historical strategy of assuming stationarity in hydrological statistics for developing water management infrastructure is no longer tenable in the face of such climatic trends. Serinaldi and Kilsby (2015), however, illustrate difficulties in using nonstationary models
for the associated hydrological frequency analysis. Future climate projections suggest that the range of hydrologic variability over many locations may move completely outside the historical ranges (Dirmeyer et al. 2016).

### 2.1.2 Examples from the Symposium

Real-world variability, including climatic trends, was addressed by several presentations at the symposium. Again, we summarize these presentations here in the form of self-contained figures, with captions detailed enough to describe the
individual studies; the captions also point the reader to relevant papers, if available, and to an appropriate contact for further information. The six figures included in this section cover a variety of topics:

- The quantification of interannual and interdecadal variability in northern Canada streamflow. The rivers studied exhibit strong interannual and interdecadal variability (see Figure 1), though no trend in total discharge during 1964-2013 (Déry et al. 2016).

- Analysis of the sources of rainfall variability over parts of Queensland, Australia (Figure 2). The variability is found to be potentially controlled more by nearby SSTs than by distant climate phenomena such as El Niño (Figure 2).

- Analysis of the impacts of model bias on the estimation of trends in discharge over the coming decades (Figure 3). Climate projection data are applied to a default land model and to a version of the model with improved (reduced bias) treatments of evapotranspiration and dynamic vegetation; the two models produce contrasting trends in streamflow associated with future drought.

- The impact of vegetation disturbance on simulated streamflow variability (Figure 4). Properly accounting for vegetation response to meteorological and hydrological variables and for feedbacks with these variables is seen to have important implications for the overall characterization of hydrological variability in a changing climate.

- Atmospheric simulation of the jet stream and atmospheric rivers (Figure 5). State-of-the-art atmospheric models are found to have an equatorward bias in their positioning of the jet stream, with consequent impacts on their simulation of atmospheric rivers and associated cold season precipitation. Improved atmospheric simulation of the jet stream may be possible with higher resolution models.

- Calibration of hydrological models with remotely sensed data (Figure 6). Globally distributed estimates of runoff generation may improve with a new computational approach keyed to certain dominant landscape processes, an approach that also permits studies of how root zone storage capacity, for example, may respond to climate variations.

Naturally, a different group of attendees would have provided a different sampling of research. This particular sampling, however, can be considered representative, indicative of the wide variety of topics now being addressed in the area of general hydroclimatic variability and trends.

## 2.2 Drought

### 2.2.1 Recent Literature

Given its societal relevance, drought has been tracked extensively in recent years. In the United States, the U.S. Drought Monitor (http://droughtmonitor.unl.edu/Home.aspx) provides a current weekly map of drought conditions, and the U.S. Seasonal Drought Outlook (http://www.cpc.ncep.noaa.gov/products/expert_assessment/sdo_summary.php) gives an indication of where drought is likely to develop or break over the coming months. The Australian Bureau of Meteorology similarly issues detailed drought statements (http://www.bom.gov.au/climate/drought/). Drought research in recent years has intensified as well, with substantial input from new measurement approaches, particularly satellite-based remote sensing. Damberg and AghaKouchak (2014), for example, utilize remotely sensed precipitation datasets to characterize recent droughts in the Northern and Southern Hemispheres. Remotely-sensed estimates of land water storage, made possible by measurements from the Gravity Recovery and Climate Experiment (GRACE) satellite, provide indications of water storage deficits that can

aid in the characterization of drought (Thomas et al. 2014). Research addressing more traditional observational sources and indices has been published as well; Sheffield et al. (2012), for example, illustrate that the traditional Palmer Drought Severity Index, based on Thornthwaite potential evaporation, may lead to overestimates of drought severity and trends.

Along with new measurement approaches come improved statistical and modeling treatments of drought, as reviewed by Mishra and Singh (2011). A Bayesian approach was recently applied by Kam et al. (2014) to connect drought probability to phases of the Atlantic Multi-decadal Oscillation (AMO), Pacific Decadal Oscillation (PDO) and El Niño / Southern Oscillation (ENSO) cycles. Pan et al. (2013) use a Copula (joint probability distribution) approach focusing on a soil moisture-based drought index and precipitation forecasts to characterize uncertainties in drought recovery. Land surface modeling in combination with observations of meteorological forcing provides a unique means for monitoring drought on the global scale (e.g., Nijssen et al., 2014). Numerical climate models have evolved substantially in the last decades, and their application to drought studies is growing; Hoerling et al. (2014), for example, use such models to analyze the 2012 United States Great Plains drought, and Coats et al. (2015) evaluate their ability to reproduce the character of paleoclimatic megadroughts in southwest North America.

The specter of climate change largely manifests itself in concerns that drought frequency will increase. Numerical model simulations of changing climate provide much of the needed data for focused study; Seager and Vecchi (2010) use these models to examine the character of future drought in southwestern North America, concluding that the occurrence of drought there can be expected to increase in the coming century due to reduced precipitation from large-scale atmospheric circulation changes during winter months. Cook et al. (2014) examine climate model simulations to quantify the relative impacts on agricultural drought of changes in precipitation and temperature (through evapotranspiration) and demonstrate that the temperature impact is substantial. Dai (2013) evaluates the historical record and climate change simulations in the context of aridity changes and concludes that the models are generally consistent with the historical record up to 2010. Regarding California drought, Mao et al. (2015) studied the historical record (rather than climate simulations) and conclude that the 2013-2014 drought was induced by reduced precipitation rather than by the observed temperatures trend, while Diffenbaugh et al. (2015) find that reduced precipitation in California is more likely during anomalously warm years. Mo and Lettenmaier (2015) find that flash drought, based on a definition of concurrent heat extreme, soil moisture deficit and evapotranspiration (ET) enhancement, has been in decline over the US during the last 100 years (though with a rebound after 2011), while recent work by Wang et al. (2016) indicates that the occurrence of flash drought in China has doubled during the past 30 years. A severe flash drought in the summer of 2013, for example, ravaged 13 provinces in southern China. Trenberth et al. (2015) highlight some of the difficulties associated with characterizing changes in drought behavior over time, pointing to deficiencies in the precipitation datasets being used and to the need to account properly for sources of natural variability, such as ENSO.

Given its importance, drought has been the subject of several recent overview and review papers; the interested reader is directed to these papers for further information. Mishra and Singh (2010) describe drought definitions and drought indices

and identify important gaps in drought research. Wood et al. (2015) provide a synthesis of research (largely focused on North American drought) performed by the National Oceanographic and Atmospheric Administration's Drought Task Force, and Schubert et al. (2016) review the latest understanding of meteorological drought as it manifests itself around the world. Kiem at al. (2016) reviews the current understanding and history of drought in the Australian context, including implications for future droughts given climate change. Peterson et al. (2013), in their overview of droughts in the United States, provide additional useful references.

### 2.2.2 Examples from the Symposium

The symposium included two presentations that focused specifically on drought mechanics and drought character:

- Drought in China (Figure 7). Drivers of seasonal (summertime) meteorological drought in northern China include the El Niño cycle and springtime Eurasian snow cover; in southern China, the probability of flash drought appears to be increasing.
- Impact of soil moisture on the atmospheric general circulation (Figure 8). Observed connections between soil moisture, clouds, convection, and subsidence may underlie a mechanism by which soil moisture influences not only local rainfall, but also the large-scale atmospheric circulation in such a way as to sustain dry anomalies from spring to summer.

Both of these presentations address mechanisms that may contribute to improved seasonal predictions of drought.

### 2.3 Floods

### 2.3.1 Recent Literature

Much recent research has addressed flash floods in Europe. Gaume et al. (2009), for example, describe their compilation of nearly 600 flash flood events in Europe, and Marchi et al. (2010) characterize European flash floods in the context of basin morphology, rainfall characteristics, antecedent soil moisture, and other factors. An extensive field experiment aimed at quantifying facets of flash floods in the northwestern Mediterranean was conducted in the fall of 2012 (Ducrocq et al., 2014). The nature of floods has been studied in other areas as well; Gochis et al. (2015) analyze the meteorological and hydrological conditions underlying the September 2013 Colorado flood event in great detail, addressing forecast capabilities and also pointing to new observations that may help prepare for future events. Berghuijs et al. (2016) examine the mechanisms underlying flood generation in the continental US and find that precipitation in isolation is not a good predictor of maximum annual flow; precipitation needs to be considered in conjunction with soil moisture and snow amounts. Teufel et al. (2016)

perform a meteorological analysis of the June 2013 Alberta floods. Huang et al. (2014) used a combination of ground-based and satellite data to map flood inundation in the Murray-Darling Basin of Australia.

Many recent studies have addressed potential changes in flood character associated with changes in climate. Mallakpour and Villarini (2015) examine the observational record in the central United States and find an increase in the frequency of flood events there, though not an increase in the largest flood peaks. Regarding future changes, Hirabayashi et al. (2013) combine climate change projections from a number of climate models with a global river routing model to determine that regions such as Southeast Asia and eastern Africa may be subject to greater flood frequency by the end of the century. Similarly, Arnell and Gosling (2016) ingest the results of climate projections from multiple climate models into a global hydrological model and, considering impacts on future distributions of human population, find indications of increased flood risk, though the magnitudes of the impacts are uncertain given the variability in the projections. Hallegatte et al. (2013) address the costs of flooding in coastal cities, which are especially prone to the effects of subsidence and sea level rise.

Hall et al. (2014), citing many recent studies, provide a thorough review of flood regime changes inferred in Europe based on observations and model experiments. Johnson et al. (2016) provide a review of historical trends and variability of floods in Australia, along with an assessment of future flood hazards given climate change. Kundzewicz et al. (2014) offer a global look at flood potential in the context of climate change and indicate a low level of confidence in current projections of the character (magnitude and frequency) of floods.

### 2.3.2 Examples from the Symposium

Several presentations at the symposium focused on floods and flooding; two are represented here:

- Flood monitoring and forecasting. A system known as the Aqueduct Global Flood Analyzer estimates flood risks across the globe, considering aspects such as flood hazard, exposure, and vulnerability (Figure 9).
- Joint analysis of flood and drought potential. Floods and droughts need to be considered together in reservoir design and operation – their joint impacts vary spatially, leading to global variations in the relative difficulty of managing hydrological variability (Figure 10).

Flood monitoring and forecasting systems are indeed important sources of information for mitigating the societal impacts of floods. The first example is one of a number of such systems described at the symposium.

**2.4 Land-Atmosphere Coupling**

**2.4.1 Recent Literature**

An important facet of climate science is the idea that the land surface is an active, dynamic component of the climate system rather than simply a passive respondent – especially the idea that soil moisture variations can imprint themselves on the overlying meteorology and on associated hydrological variability. Seneviratne et al. (2010) provide an extensive overview of research into the nature of this land-atmosphere coupling. The continuing research is shedding new light on the ability of soil moisture to influence, for example, rain variability and heat waves.

The soil moisture-air temperature connection is intuitive; drier soils evaporate less and thus experience less evaporative cooling, leading to higher temperatures for the local system. This connection has been examined, for example, in the context of the 2003 European heat wave (Fischer et al. 2007). More difficult to pin down is the soil moisture-precipitation connection. Indeed, the literature indicates complexities regarding the directions of the feedback, i.e. in whether increased soil moisture leads to increased or decreased rainfall. For example, Findell et al. (2011) find that over the eastern United States, increased soil moisture leads to a greater probability of afternoon rainfall, supporting the idea of positive feedback, whereas Taylor et al. (2012) provide observational evidence that rainfall tends to fall over the drier patches in a landscape. Guillod et al. (2015) address the apparent contradiction by showing that large-scale wet conditions are in general favorable to increased precipitation (a positive temporal correlation at the large scale), yet rainfall can favor the drier patches within the broadly wet conditions (a negative spatial correlation). Theory suggests that some atmospheric conditions promote a positive soil moisture-rainfall feedback whereas others promote a negative one; Ferguson and Wood (2011), through an analysis of satellite-based data, separate the globe into the associated different coupling regimes, and Roundy et al. (2013) extend the methodology to show how the coupling regime in a given location can change with time.

Naturally, land-atmosphere coupling has been studied extensively within climate models. One recent study (Saini et al., 2016) examines past drought events using a regional climate model with different soil moisture initializations; soil moisture feedback is found to be much more important for the development of the 2012 drought in the central U.S. than for the development of the 1988 drought there, due to the lack in 2012 of a clear large scale forcing favoring drought. Using a different model, Koster et al. (2016) show that soil moisture deficits in the interior of North America can help generate atmospheric circulation patterns that in turn can contribute to the persistence and areal expansion of the dryness. Regarding the impact of climate change on land-atmosphere coupling, Dirmeyer et al. (2013a,b, 2014b) analyze the water cycle in CMIP5 models in several ways, noting evidence for enhanced land-atmosphere feedbacks in a changing climate arising in concert with increasing extremes. Worth noting, though, is that models with parameterized convection may have difficulty in properly representing land-atmosphere coupling. Recent advances in convection-permitting modeling may lead to better simulations of convection and land-atmosphere interactions (e.g., Hohennegger et al. 2009; Leung and Gao 2016).

Some recent work has advocated a more holistic treatment of land-atmosphere coupling, one that considers the co-evolution of snow properties, cloud forcing, temperature, relative humidity, precipitation, wind, and boundary layer growth. On the Canadian Prairies, for example, the monthly variability of temperature and relative humidity in the warm season is dominated by shortwave cloud forcing, and as a result, both equivalent potential temperature and the lifting condensation level, which drive moist convective development, depend strongly on cloud forcing (Betts et al. 2013a, 2015, 2016). This has implications for seasonal predictability, given the uncertainties in predicting daily cloud forcing in numerical forecast models. Betts et al. (2017) provide a set of coupling coefficients between the near-surface diurnal cycle of the moist thermodynamic variables, cloud forcing and lagged precipitation for model evaluation. Another challenge for seasonal predictability is the dynamic coupling between vegetation phenology, precipitation anomalies, soil water extraction, and evapotranspiration. The intensification of cropping increases evapotranspiration and cools the summer climate both in the US Midwest (Mueller et al. 2016) and the Canadian Prairies (Betts et al. 2013b), and the extraction of soil water during the growing season appears to dampen precipitation anomalies (Betts et al. 2014b) and perhaps contributed to the onset of the 2012 Great Plains drought (Sun et al. 2015).

The Global Land Atmosphere System Study (GLASS) panel of the Global Energy and Water Exchanges (GEWEX) project has focused recently on the definition and evaluation of land-atmosphere coupling processes in models and observational data (Santanello et al. 2011) with a particular focus on the hydrologic cycle. The reader is directed to the website http://cola.gmu.edu/dirmeyer/Coupling_metrics.html for an evolving summary of land-atmosphere coupling metrics and associated references.

### 2.4.2 Examples from the Symposium

Symposium papers addressed several facets of land-atmosphere coupling, including the attribution of the sources of the coupling strength simulated by an Earth system model and the evaluation of simulated coupling characteristics with relevant observational datasets. One of these presentations is represented here:

- Joint analysis of surface and boundary layer data (Figure 11). The analysis of an extensive dataset collected over the Canadian Prairies, in the context of the aforementioned holistic approach to analyzing land-atmosphere interaction, reveals important connections between cloud radiative forcing and near-surface air temperature, including how these connections change in the presence of snow cover.

**2.5 Hydrological prediction**

**2.5.1 Recent Literature**

Again, a key motivation for studying hydroclimatic variability is improvement in the skill of hydrological predictions – skillful predictions can allow society to prepare itself better for upcoming hydrological variations. One highly relevant tool for this is the extended-range forecast system, a coupled ocean-atmosphere-land modeling system that provides, among other things, forecasts of temperature and rainfall over continents weeks to months in advance. Doblas-Reyes et al. (2013) provide a review of the state-of-the-art in seasonal forecasting with such systems, Yuan et al. (2015) provide a review of climate model-based seasonal hydrological forecasting, and Robertson et al. (2015) and Vitart et al. (2017) describe emerging operational subseasonal-to-seasonal (S2S) forecast systems. Regarding the overall accuracy of seasonal forecasts, Roundy and Wood (2015) use statistical models to examine how such forecasts may be limited by biases in their treatment of land-atmosphere coupling, and Yuan and Wood (2012) address critical questions regarding the combination of forecasts from different systems – whether redundancies amongst the systems can be properly accounted for when developing a multi-model forecast.

In essence, forecast skill in a subseasonal-to-seasonal forecast system is derived from the information content inherent in the system's initialization. Therefore, considerable effort has been directed toward improving this initialization, for example, through the improvement of Bayesian (Kalman and particle filters) and variational (1D-4D) data assimilation methods as applied to the initialization of high-dimensional models (e.g. Li et al. 2015; van Leeuwen, 2015). A promising strategy is based on combining advantageous characteristics of both paradigms (e.g., the probabilistic estimates for Bayesian methods and the broader evaluation window for variational ones), as demonstrated by, for example, Bruehner et al. (2010) and Noh et al. (2011).

While the initialization of ocean states has long been considered key for the coupled forecast systems (NRC, 2010), there is growing recognition that the initialization of various land states may be just as critical to extracting otherwise unattainable facets of skill (e.g., Dirmeyer and Halder 2017). Soil moisture impacts on subseasonal forecast skill is quantified across a broad range of systems in the Global Land-Atmosphere Coupling Project (Koster et al. 2011; van den Hurk et al. 2011); impacts are found to be much larger on temperature forecast skill, but impacts on precipitation forecast skill are significant in places, particularly when considering the strongest initial soil moisture anomalies. A positive impact of snow initialization on seasonal temperature forecast skill is demonstrated by Peings et al. (2011) and Lin et al. (2016); the latter show that the assimilation of satellite measurements improves the initialization, with concomitant impacts on the forecast skill. Koster and Walker (2015) show that when a dynamic plant phenology model is used in a forecast system, initializing the vegetation state (e.g., the leaf area index) has a positive impact on temperature forecasts but not on precipitation forecasts. Subsurface temperature is another variable to consider; Xue et al. (2016) demonstrate that initializing these temperatures in an atmospheric modeling system can improve the simulation of subsequent drought. As shown by Dirmeyer et al. (2013c), the predictability

of meteorological variables (the theoretical maximum forecast skill that can be derived from an initialization) may change as the climate changes.

Drought forecasting in particular has been a focus of much recent work. In sub-Saharan Africa, an advanced drought monitoring and forecasting system based on hydrological modeling, remote sensing, and seasonal forecasts has been developed and implemented, for example, at regional weather and climate centers in Niger and Kenya (Sheffield et al., 2014). Regarding the skill of seasonal drought forecasts, results are mixed. Yuan and Wood (2013), in an analysis of multiple seasonal forecast systems, uncover significant limitations in the ability of such systems to forecast drought. Quan et al. (2012), however, using a specific seasonal forecast system, demonstrate that the sea surface temperatures produced in the system, particularly those associated with El Niño cycles, add some skill to drought prediction over the United States. Roundy et al. (2014) demonstrate that apparent deficiencies in the simulated land-atmosphere coupling behavior of a forecast system can limit its ability to predict and maintain drought.

Streamflow forecasting has obvious relevance to water resources management, and relative to drought forecasting, it can rely less on dynamical seasonal forecasts given the strong connection between streamflow and, for example, snow storage at the start of a forecast period. Koster et al. (2010) and Mahanama et al. (2012), without using a dynamical forecast model, produce accurate streamflow forecasts at seasonal lead times based solely on initial snow and soil moisture information. This said, seasonal climate forecasts (perhaps combined with medium-range weather forecasts, as described by Yuan et al. [2014]) can add skill to long-term streamflow forecasts (Yuan et al. 2013).

Demargne et al. (2014) describe in detail the operational Hydrologic Ensemble Forecast Service, which provides, through integration of multiple inputs (including meteorological forecasts), streamflow forecasts at leads from 6 hours to 1 year. Pagano et al. (2014) outline the challenges faced by forecast agencies around the world in developing an operational river forecasting system that is suitably effective.

**2.5.2 Examples from the Symposium**

Symposium presentations focusing on hydrological prediction and forecasts include analyses of:

- Data assimilation approaches for forecast initialization (Figure 12). A data assimilation approach called OPTIMISTS (Optimized PareTo Inverse Modeling through Integrated Stochastic Search) combines features from Bayesian and variational methods for the initialization of highly distributed hydrological models (Figure 12).
- Land-atmosphere coupling strength in a forecast model (Figure 13). The idea that the US operational forecast model underestimates land-atmosphere coupling is inferred from the fact that observed precipitation rates are more closely related to antecedent soil moisture than are model simulated rates.

Of course, improved hydrological prediction is an "end goal" of much of today's hydrological research. Prediction is thus an important sub-theme of many of the other examples provided in this paper.

## 3. Summary and Outlook

The present paper provides an overview of some recent research (roughly since 2010) on the subject of hydrological variability and predictability, with particular focus on the spatial as well as temporal aspects of variability and with an eye toward large-scale prediction. Given the wealth of research on the subject, this overview does not pretend to be comprehensive, even for the recent period; it is perhaps best considered a starting point for those interested in pursuing this multi-faceted topic further. The specific examples shown in the figures were culled from relevant presentations made at the Symposium in Honor of Eric
Wood: Observations and Modeling across Scales. These examples are representative of the breadth of today's research on this topic.

Together, this literature survey and the figures demonstrate that this is a unique period in the hydrological sciences for at least two reasons. First, on the positive side, hydrologists now have access to powerful new analysis tools and to unprecedented global datasets, and they have a deeper appreciation of the global nature of the hydrological cycle and its connections to the
rest of the Earth system. Improvements in hydrological tools is exemplified by the growing complexity of numerical hydrological models in terms of both resolution and their treatments of critical hydrological processes – such models can serve as invaluable laboratories for hydrological analysis. Hydrological data availability has been revolutionized by remote sensing data, which can provide global information on soil moisture, precipitation, vegetation health, and so on; in situ observational networks are also providing large-scale pictures of critical hydrological fields. Combining the complex models with the
unprecedented data coverage and with enhanced analysis techniques (such as improved data assimilation strategies) indeed sets the stage for improved hydrological prediction at the large scale. Such prediction efforts, which are often performed in the context of Earth system models, exemplify the growing appreciation of the importance of large-scale hydrology – the importance of addressing aspects of the science that extend beyond traditional catchment boundaries.

On the negative side, daunting hydrology-related challenges to society are becoming ever more prominent. Global increases in population are leading to increased water demand, and at the same time, reduced levels of water quality (due to pollution, saltwater intrusion, etc.) are reducing water availability. To some extent the ever-shrinking buffer between water supply and water demand can be addressed by improvements in hydrological prediction at multiple time scales (weather through decadal), given that the overall efficiency of water usage would necessarily benefit from foreknowledge of specific variations and trends
in water availability. Floods and droughts represent extremes in water supply variations, and their improved prediction would not only improve the efficiency of water usage but also mitigate tremendous economic losses associated with crop failures and

damage to infrastructure. Note that all of the pressing societal needs requiring improved hydrological understanding and prediction come against the backdrop of potential nonstationarities associated with anthropogenic climate change, nonstationarities that may eventually lead, at least on regional scales, to greater deficiencies of water availability relative to demand.

Such challenges can only be addressed with continued hydrological research, of the type surveyed in this paper. Given these challenges, and given the growing availability of powerful tools and datasets to address them, large-scale, climate-oriented hydrological variability studies will undoubtedly continue to be a vibrant component of Earth system science.

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

**Figure Captions**

Figure 1: Climatic change may manifest itself as changes in the statistics of streamflow, and such changes can have important implications for water resource management. A recent study searched for trends in the streamflow within six basins of Northern Canada; results are shown above. Each box represents a specific basin: a) Bering Sea; b) Western Arctic Ocean; c) Western Hudson and James Bay; d) Eastern Hudson and James Bay; e) Eastern Arctic Ocean (Hudson Strait/Ungava Bay); and f) Labrador Sea. Within each basin, after determining a mean and standard deviation from the 50 years of data, the flow for each year was standardized, and the average standardized streamflow for each decade of interest (1964-1973, 1974-1983, 1984-1993, 1994-2003, and 2004-2013) was computed and plotted above as a red square. Similarly, the coefficient of variation of total river discharge for each decade was computed from the mean and standard deviation of discharge within that decade and plotted as a blue circle. (Note that values of the coefficient of variation have been multiplied by 10.) The streamflow amounts in the different basins clearly show strong decadal variability; however, they lack a clear trend. [Contact: Stephen Déry.]

Figure 2. It is widely assumed that large-scale SST patterns (the El Niño / La Niña pattern, for example) have an important impact on rainfall variability in regions like Australia. More proximate SSTs, however, may be just as important. This was investigated through a comparison of two 40-member ensembles of WRF (regional model) simulations, the first using observed SSTs and the second using SSTs associated with previous La Niña events. Both ensembles employed the same atmospheric forcing along the WRF model's lateral boundary. Shown in the plot is the inferred contribution of local SSTs to the major flooding that occurred between 10 and 20 December 2010 in Queensland, Australia. In many places the high local SSTs (within a few hundred km of the coast) accounted for more of the precipitation than did the prevailing La Niña conditions, at least at the spatial scales considered here. The analysis demonstrates limitations in hydrological predictability based solely on large-scale climate modes such as El Niño /

La Niña. Controls on hydrological variability and predictability are in fact more complex. [Contact: Jason Evans. See Evans and Boyer-Souchet (2012) for further information.]

Figure 3. Changes in climate in the coming decades will presumably be accompanied by changes in hydrological behavior at the Earth's surface – changes in the character, for example, of streamflow. Our estimates of such changes, however, may be severely limited by biases in the models used to quantify them. This is demonstrated here with two simulations of hydrological behavior in the Connecticut River Basin, one using the default VIC model and the other using a version of VIC with bias-corrected evapotranspiration (VICET). The VICET model overwrites the model-estimated ET components from VIC with bias-corrected values, and such correction propagates to improve the estimation of other hydrological variables. The meteorological forcing for the two simulations is identical, which for the historical segment was derived from NLDAS-2 (Xia et al., 2012) and for the future segment was constructed based on bias correction of the NARCCAP projection following the approach of Ahmed et al. (2013) using NLDAS-2 as the observational reference. Shown in the plot, for each simulation and for both time periods, are the 5-day minimum discharges at the Thompsonville Station (in cfs). The strong model dependence in the hydrological projections indicate a strong need for careful evaluation and improvement of land model parameterizations. [Contact: Guiling Wang. See Parr et al. (2015) for further information.]

Figure 4. The characterization of hydrological changes associated with climate change requires a consideration of vegetation disturbance, as indicated by a number of simulations of San Juan River basin streamflow with the Variable Infiltration Capacity (VIC) model. Several simulations are considered here: one using historical (1970-1999) meteorological forcing (average streamflow shown as a thick black line) and others using future (2070-2099) temperature and precipitation forcing from the IPCC's CMIP5 database (four different sets of forcing from four different Earth System Models, or ESMs). Future streamflow conditions are provided for two vegetation disturbance scenarios. The thin black line (with gray shading underneath) represents the average seasonal cycle of simulated streamflow from future runs which utilize the historical representation of vegetation. The green envelope (mean is shown as a dashed green

line), on the other hand, represents the range of average seasonal cycles produced in future runs (one for each of the 4 ESMs) that results from the imposed forest mortality of close to 90% by the 2080s, based on work from McDowell et al. (2016). We see that for the San Juan River basin, a major tributary to the Colorado River basin, spring freshet in the future runs occurs earlier in the season, shifting from mid-May to the end of April. Flows are projected to be higher during late fall, winter and early spring, and lower during late spring, summer and early fall. Disturbing the vegetation in addition to using projected temperature and precipitation forcing results in a different pattern of streamflow, with lower flows in early spring and then higher peakflow, and with lower recessional summer flows due to differences in how regrowth vegetation (i.e. shrubs) partitions water and snowpack. Studies on climate change thus require a consideration of changes in vegetation dynamics; otherwise results may be misleading or could underestimate impacts (Bennett et al. in review). [Contact: Katrina Bennett]

Figure 5. Atmospheric rivers (ARs) are responsible for over 90% of the moisture transport to the extratropics (Zhu and Newell 1998). They also contribute significantly to heavy precipitation and flooding in many regions worldwide (Ralph et al. 2006). Understanding how ARs may change in a warmer climate is important for managing water resources and flood risk. Associated with Rossby wave breaking, the frequency of ARs and their landfall locations are influenced by the jet stream. Global climate models in the Coupled Model Intercomparison Phase 5 (CMIP5) exhibit an equatorward bias in the simulated jet position. For example, the left panel shows the grid boxes (colored) used to detect CMIP5 model-simulated North Atlantic ARs making landfall in Europe. The black and blue horizontal lines show the CMIP5 and reanalysis mean jet positions, respectively. The CMIP5 models simulate a mean jet stream position that is almost 5° equatorward of that depicted in the reanalysis, probably due to their relatively coarse model resolutions (e.g., Lu et al. 2015). Biases in the jet position have important implications for the simulation of ARs in Europe. As shown in the right panel, CMIP5 models simulated too few (too many) ARs poleward (equatorward) of the observed jet position in the North Atlantic during December-February compared to four global reanalyses (color symbols). Here, the box-and-whisker plots show the CMIP5 multi-model mean (dot), median (horizontal bar), 75% and 25% percentiles (upper and lower boundaries of the box), and the highest and lowest values (whiskers). A challenge for improving the

simulation of ARs and their response to warming is the more accurate simulation of the jet stream and the associated

Rossby wave dynamics. Enabled by advances in computational resources, increasing model resolution may improve

the fidelity of model simulated jet, which may improve projections of changes in extreme precipitation and flooding

in a changing climate. [Contact: Ruby Leung.  See from Gao et al. (2016) for more information.]

Figure 6.   Readily available remote sensing products can be used to constrain hydrological models in a way that allows

streamflow prediction in ungauged basins.  The above schematic shows the relevant connections to consider during

a calibration procedure.  HAND refers to the Height Above the Nearest Drainage (which is the hydraulic head), root

zone storage capacity is the maximum amount of soil water that can be accessed by the vegetation root systems, and

the recession time scale parameter controls the steepness of the recession.  P, E, and W represent precipitation,

evaporation, and soil water content, with RS indicating a remotely sensed source.  $S_{u,max}$ is the root zone storage

capacity, $K_s$ is the slow recession time scale, and β, D, and $K_f$ are the exponent of the threshold function for runoff

generation, the splitter between recharge and runoff, and the fast recession time scale, respectively.  Note that the root

zone storage capacity of ecosystems reflects in part the ability of vegetation to distribute its roots to optimize soil

water usage.  Through the calibration scheme shown above, we can use historical time series of precipitation and

evaporation to derive the effective storage capacity utilized by the ecosystem and then connect it to the ecosystem's

survival strategy (Gao et al., 2014).  In addition, through such an approach, we can investigate how ecosystems will

adjust their storage capacity in response to climatic change and how rainfall-runoff relations will change as a result.

[Contact: Hubert Savenije.  See Savenije and Hrachowitz (2016, 2017) for more information.]

**Figure 7.** Joint analysis of a variety of climate variables provides new insights into the predictability of seasonal drought in

China and into recent changes in the character of flash drought there.  The top panels show (a) the slopes (in

geopotential meters, or gpm) of the regressions of July-August 500 hPa geopotential height anomaly on detrended

(and standardized) July NINO3.4 index and (b) the slopes (also in gpm) of the regressions of this height anomaly on

negative (and standardized) March Eurasian snow cover.  The two panels demonstrate that both ENSO and Eurasian

snow cover are statistically tied to the Eurasia teleconnection (EU) pattern responsible for summer droughts in northern China (modified from Wang et al., 2017). Note that a seasonal climate forecast model usually shows higher forecast skill during ENSO years; the CFSv2 model, for example, predicted the 2015/16 El Niño and roughly captured the devastating North China drought in the summer of 2015.  However, a strong El Niño does not necessarily result in an extreme drought in North China, since such drought also depends on whether the El Niño evolves synergistically with Eurasian spring snow cover reduction to trigger a positive summer Eurasian teleconnection (EU) pattern (a-b) that favors anomalous northerly air sinking over North China (see Wang et al. 2017 for more information). Regarding changes in the character of flash drought, the two bottom panels show (c) changes in flash drought events (events per year) over southern China and (d) changes in standardized (and thus dimensionless) precipitation and surface air temperature averaged over southern China. The increasing trend in flash drought over southern China suggests that the probability of concurrent heat extremes, soil moisture deficits, and positive evapotranspiration anomalies there is increasing (see Wang et al., 2016 for more information). [Contact: Xing Yuan]

Figure 8.  The possibility that soil moisture anomalies can affect the character of the overlying atmospheric circulation could have profound implications for our understanding of drought evolution and maintenance.  The plot above shows the statistical connection between soil moisture (as derived from offline land analyses) and 500 hPa geopotential height anomalies (as derived from an atmospheric reanalysis).  More specifically, the red curve shows the lead-lag correlation between pentad soil moisture anomalies and the height anomalies during May-July (MJJ) over the south-central United States over the period 1981-2012, whereas the blue line depicts the autocorrelation function (ACF) of the pentad 500 hPa geopotential height anomalies of MJJ for the same region and period.  The ACF values have been multiplied by -1 for easy comparison with the red curve.  The 95% confidence bounds are derived as the standard deviations divided by the square roots of N, where N is the effective number of independent samples.  (The original sample size is n=612, whereas N=139 after accounting for autocorrelation in the time series.)  The fact that the red curve lies below the blue curve (and is significant) for -1 to -6 pentads indicates that positive large-scale mid-tropospheric geopotential height anomalies (which are characteristic of circulation patterns associated with drought)

are more correlated with soil moisture deficits 5-30 days earlier than they are with earlier height anomalies, suggesting that the patterns may be influenced more by soil moisture than by the memory of the large-scale atmospheric circulation (either remotely forced by SSTA or through memory provided by the internal atmospheric variability). This result provides observational evidence of soil moisture feedback on large-scale drought circulation in summer over the south central US (or southern Plains). [Contact: Rong Fu. Figure taken from Fernando et al. (2016); see this reference for more information.]

Figure 9. Scientific progress in conjunction with advances in web-based software technologies are providing society with valuable new tools for coping with the physical and economic uncertainties associated with flooding. The above screenshot, for example, is from the Aqueduct Global Flood Analyzer, a web-based interactive platform that estimates river flood risk in terms of urban damage, affected GDP, and affected population at the country, state, and river basin scale across the globe. The Analyzer enables users to estimate current flood risk for a specific geographic unit, taking into account existing local flood protection levels. It also allows users to project future flood risk under climate and socio-economic change and separately attribute change in flood risk to each of these drivers. Finally, for each flood protection level, high-resolution maps of yearly flooding probability are provided. The basis for the Analyzer is the global hydrology and water resources model PCR-GLOBWB (Van Beek et al., 2011). The methodology behind the tool is described extensively in Ward et al. (2013) and Winsemius et al. (2015). Current developments for this tool entail adding the risk of coastal flooding and analyzing the costs and benefits of adaptation measures, including traditional "hard defenses" and nature-based solutions. (Adapted from Bierkens [2015]. Contact: Marc Bierkens)

Figure 10: In nature, changes in the storage of water in a hydrological basin can smooth out hydrological variations associated with floods and droughts. The spatial variability in necessary hydrological storage, however, remains relatively unstudied – at the present time there is no global map showing the storage needed to ameliorate floods and droughts, either for the present climate or under climate change. In the panels above, using the Ganges-Brahmaputra-Meghna basin as an example, the needed storage at each grid cell within the basin is calculated with a new method: intensity-

duration-frequency curves of flood and drought (flood duration curve and drought duration curve: FDC-DDC, an alternative representation of discharge time series obtained from a calibrated hydrological model called BTOPMC – see Takeuchi and Masood, 2016). For simplicity, the target release ($Q_T$) for smoothing is assumed to be the long term mean discharge ($Q_{mean}$) at each grid cell (Takeuchi and Masood, 2016). The figure shows a typical FDC-DDC

curve for a grid cell and an illustration of how to calculate necessary storage (top left), the spatial distribution of storage (in units of km$^3$) needed to smooth floods in the basin (bottom left), and the spatial distribution of storage (in units of months) needed to smooth flood (top right) and drought (bottom right). Note that storages expressed in months, calculated by dividing the necessary storage volume by the local $Q_{mean}$ for 1979-2003, provide a unique perspective on storage requirements. The geographical distribution of necessary storage reflects hydrological

heterogeneity associated with meteorological inputs, topography, geology, soil, vegetation, landuse, and so on. Quantifying the relationships between spatially distributed necessary storages and the geographical distribution of hydroclimatological, geological and land cover conditions can lead to improved hydrological analysis and produce useful information for water resources managers. (Contact: Muhammad Masood )

Figure 11. Land surface hydrological processes and atmospheric (boundary layer) processes do not proceed in isolation from each other; land states and boundary layer states evolve together, as a joint system. The nature of this coupled system was recently elucidated through a careful analysis of a wealth of land surface and boundary layer data collected by trained observers in the Canadian Prairies. These observers recorded hourly, since 1953, the fraction of the sky covered by opaque reflective cloud, providing daily shortwave and long-wave cloud forcing (SWCF and LWCF) on

climate timescales when calibrated against baseline surface radiation measurements (Betts et al. 2015). The panels above express some of the important relationships inherent in these data in the form of average diurnal temperature cycles for January (top left), July (bottom left), and the fall transition month of November (bottom right). For each month, days are binned by daily mean opaque cloud fraction in tenths, with a different color scheme for cold days with mean temperature <0$^o$C and snow cover, and days >0$^o$C and no snow cover. In July, the diurnal cycle of

temperature and relative humidity is dominated by SWCF on both daily and monthly timescales, and temperatures

rise under clear skies. In contrast, in January the temperatures are lower under clear skies as LWCF dominates (Betts et al. 2014a, 2015).  It is in fact the presence or absence of reflective snow cover that determines the impact of clouds on surface temperature – in November, the snow-free days are more than 10K warmer than the snow-covered days, and the former shows the July type of behavior whereas the latter shows the January type of behavior.  [Contact: Alan
Betts.  Adapted from Betts and Tawfik (2016).]

Figure 12. The success of hydrological prediction depends largely on the accuracy of the initialization of the forecast model. Advanced mathematical tools (i.e., data assimilation algorithms) are now available to transform a given set of observations into the best forecast initialization possible.  The table above outlines the features of three data
assimilation approaches: standard Bayesian data assimilation algorithms (KF stands for Kalman Filter, EnKF stands for Ensemble Kalman Filter, and PF stands for Particle Filter), variational methods, and a new technique – OPTIMISTS – that combines the advantageous characteristics of the first two.  Some of the features selected for OPTIMISTS, such as non-Gaussian probabilistic estimation and support for non-linear model dynamics, are considered advantageous in the literature (van Leeuwen, 2015); flexible configurations are available for other features
(e.g., the choice of optimization objectives or the analysis time step) for which no consensus has formed.  In the bottom panel, different configurations of OPTIMISTS (indicated along x-axis) are compared in terms of their success in improving streamflow forecasts.  The experiments were conducted with the Distributed Hydrology Soil Vegetation model (DHSVM) on a test case with 1,472 cells and over 30,000 state variables; the ordinate shows the change, relative to a control that uses no data assimilation, in the Nash-Sutcliffe Efficiency (NSE) coefficient (positive values
indicating forecast skill improvement). Asterisks on the boxplots indicate outliers. Three configurations of OPTIMISTS provide statistically significant advantages (demonstrated by the indicated p-values from the ANalysis Of VAriance): (i) setting the analysis time step equal to the entire two-week assimilation period; (ii) maximizing the consistency of the states with the background (and not only minimizing the error); and (iii) using only Bayesian sampling to generate new members/particles. Studies like this are critical for maximizing the effectiveness of the

techniques used to initialize forecast models; this particular study positions OPTIMISTS as a capable and flexible framework. [Contact: Xu Liang.]

Figure 13.  If, in the real world, land surface variations (e.g., in soil moisture) are able to affect the overlying atmosphere, and if an atmospheric model does not capture adequately this land-atmosphere feedback, the performance of the model will suffer.  A forecast model that lacks this feedback likely cannot translate the information contained in soil moisture states into improved forecasts of air temperature and precipitation.  With this as motivation, the panels above provide an evaluation of land-atmosphere feedback in the US operational forecast model (CFSv2).  The three columns show from left to right the pair-wise correlations (i) between monthly CFSv2 reforecast precipitation ($P_{CFS}$) and observed precipitation ($P_{Obs}$), (ii) between $P_{CFS}$ and reforecast initial soil moisture in layer 2 (10-40cm depth; $SM_{IC}$), and (iii) between $P_{Obs}$ and $SM_{IC}$, all for forecasts validating during JJA.  The rows show the different leads (in days) considered. Dark colors (beyond ±0.11) are significant at the 95% confidence level. The fact that observed precipitation rates are more closely related to antecedent soil moisture than are model simulated rates suggests that the US operational forecast model underestimates land-atmosphere coupling. An improvement in the system's simulation of coupled land-atmosphere processes could improve the accuracy of the forecasts produced. [Contact: Paul Dirmeyer.  Figure taken from Dirmeyer (2013); see this reference for further information]

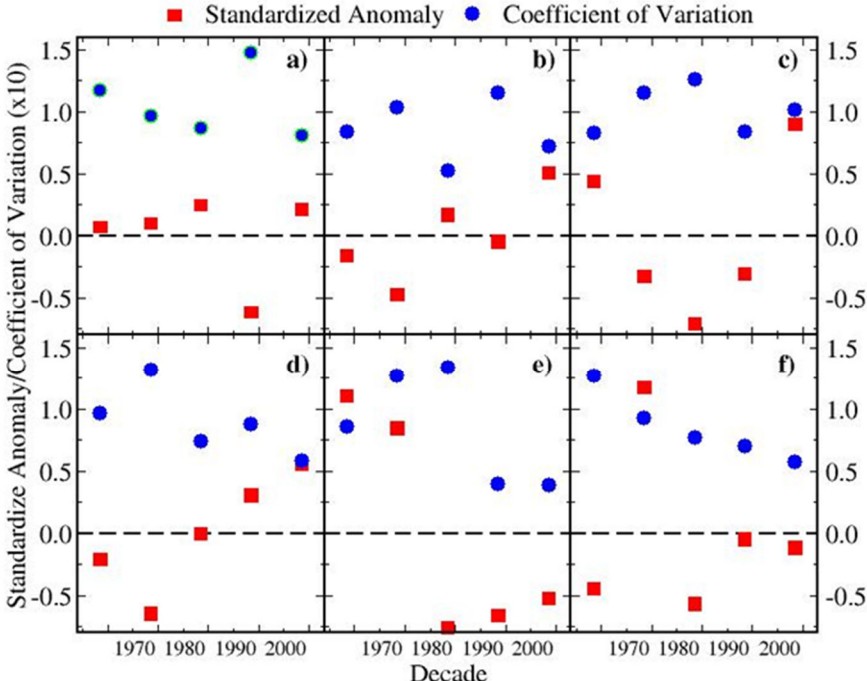

Figure 1: Climatic change may manifest itself as changes in the statistics of streamflow, and such changes
can have important implications for water resource management.  A recent study searched for trends in
the streamflow within six basins of Northern Canada; results are shown above.  Each box represents a
specific basin: a) Bering Sea; b) Western Arctic Ocean; c) Western Hudson and James Bay; d) Eastern
Hudson and James Bay; e) Eastern Arctic Ocean (Hudson Strait/Ungava Bay); and f) Labrador Sea.
Within each basin, after determining a mean and standard deviation from the 50 years of data, the flow
for each year was standardized, and the average standardized streamflow for each decade of interest
(1964-1973, 1974-1983, 1984-1993, 1994-2003, and 2004-2013) was computed and plotted above as a
red square.  Similarly, the coefficient of variation of total river discharge for each decade was computed
from the mean and standard deviation of discharge within that decade and plotted as a blue circle.  (Note
that values of the coefficient of variation have been multiplied by 10.)  The streamflow amounts in the
different basins clearly show strong decadal variability; however, they lack a clear trend.  [Contact:
Stephen Déry.]

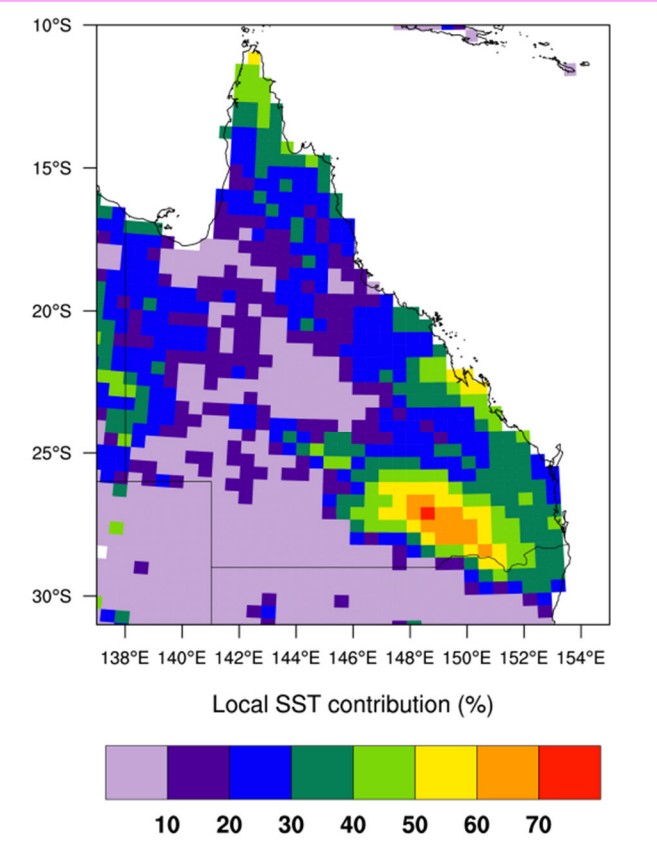

Local SST contribution (%)

Figure 2.  It is widely assumed that large-scale SST patterns (the El Niño / La Niña pattern, for example) have an important impact on rainfall variability in regions like Australia.  More proximate SSTs, however, may be just as important.  This was investigated through a comparison of two 40-member ensembles of WRF (regional model) simulations, the first using observed SSTs and the second using SSTs associated with previous La Niña events.  Both ensembles employed the same atmospheric forcing along the WRF model's lateral boundary.  Shown in the plot is the inferred contribution of local SSTs to the major flooding that occurred between 10 and 20 December 2010 in Queensland, Australia. In many places the high local SSTs (within a few hundred km of the coast) accounted for more of the precipitation than did the prevailing La Niña conditions, at least at the spatial scales considered here. The analysis demonstrates limitations in hydrological predictability based solely on large-scale climate modes such as El Niño / La Niña.  Controls on hydrological variability and predictability are in fact more complex. [Contact: Jason Evans.  See Evans and Boyer-Souchet (2012) for further information.]

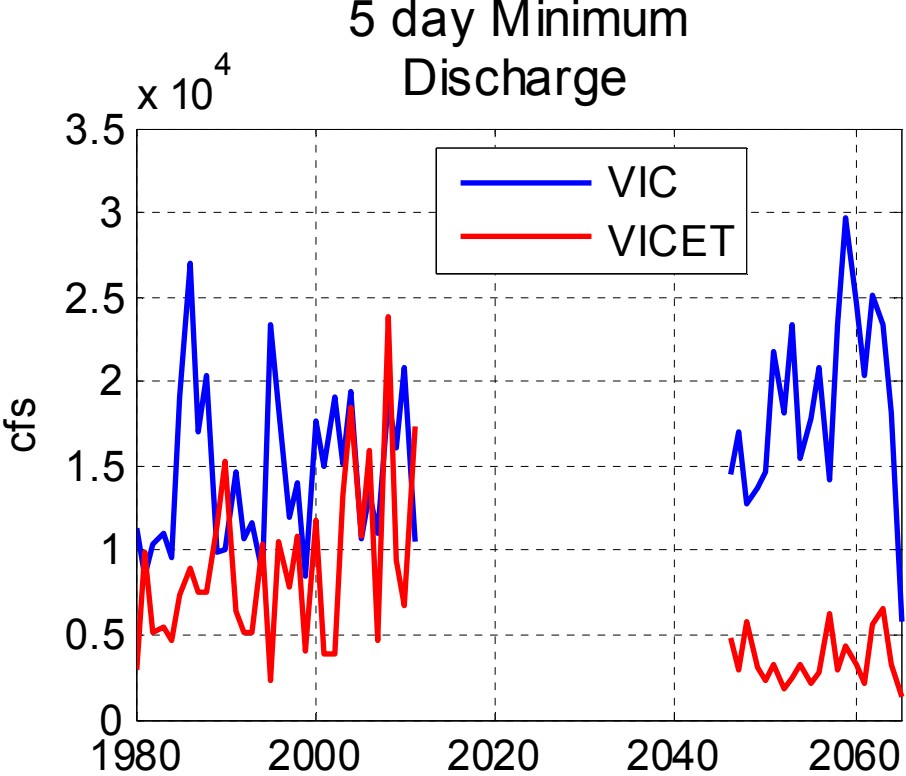

Figure 3. Changes in climate in the coming decades will presumably be accompanied by changes in hydrological behavior at the Earth's surface – changes in the character, for example, of streamflow. Our estimates of such changes, however, may be severely limited by biases in the models used to quantify them. This is demonstrated here with two simulations of hydrological behavior in the Connecticut River Basin, one using the default VIC model and the other using a version of VIC with bias-corrected evapotranspiration (VICET). The VICET model overwrites the model-estimated ET components from VIC with bias-corrected values, and such correction propagates to improve the estimation of other hydrological variables. The meteorological forcing for the two simulations is identical, which for the historical segment was derived from NLDAS-2 (Xia et al., 2012) and for the future segment was constructed based on bias correction of the NARCCAP projection following the approach of Ahmed et al. (2013) using NLDAS-2 as the observational reference. Shown in the plot, for each simulation and for both time periods, are the 5-day minimum discharges at the Thompsonville Station (in cfs). The strong model dependence in the hydrological projections indicate a strong need for careful evaluation and improvement of land model parameterizations. [Contact: Guiling Wang. See Parr et al. (2015) for further information.]

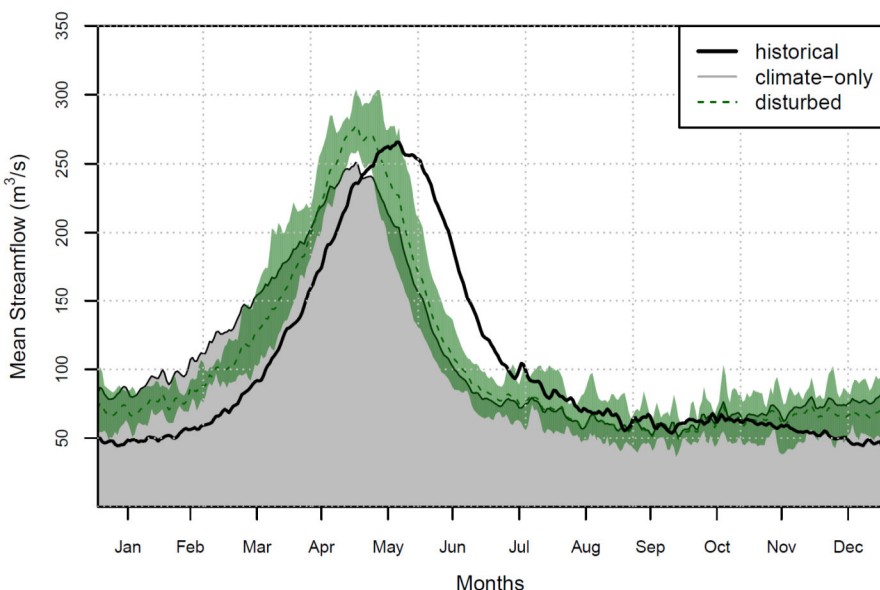

Figure 4.   The characterization of hydrological changes associated with climate change requires a
consideration of vegetation disturbance, as indicated by a number of simulations of San Juan River basin
streamflow with the Variable Infiltration Capacity (VIC) model.  Several simulations are considered here:
one using historical (1970-1999) meteorological forcing (average streamflow shown as a thick black line)
and others using future (2070-2099) temperature and precipitation forcing from the IPCC's CMIP5
database (four different sets of forcing from four different Earth System Models, or ESMs).   Future
streamflow conditions are provided for two vegetation disturbance scenarios.  The thin black line (with
gray shading underneath) represents the average seasonal cycle of simulated streamflow from future runs
which utilize the historical representation of vegetation. The green envelope (mean is shown as a dashed
green line),  on the other hand, represents the range of average seasonal cycles produced in future runs
(one for each of the 4 ESMs) that results from the imposed forest mortality of close to 90% by the 2080s,
based on work from McDowell et al. (2016). We see that for the San Juan River basin, a major tributary
to the Colorado River basin, spring freshet in the future runs occurs earlier in the season, shifting from
mid-May to the end of April. Flows are projected to be higher during late fall, winter and early spring,
and lower during late spring, summer and early fall. Disturbing the vegetation in addition to using
projected temperature and precipitation forcing results in a different pattern of streamflow, with lower
flows in early spring and then higher peakflow, and with lower recessional summer flows due to
differences in how regrowth vegetation (i.e. shrubs) partitions water and snowpack. Studies on climate
change thus require a consideration of changes in vegetation dynamics; otherwise results may be
misleading or could underestimate impacts (Bennett et al. in review).  [Contact: Katrina Bennett]

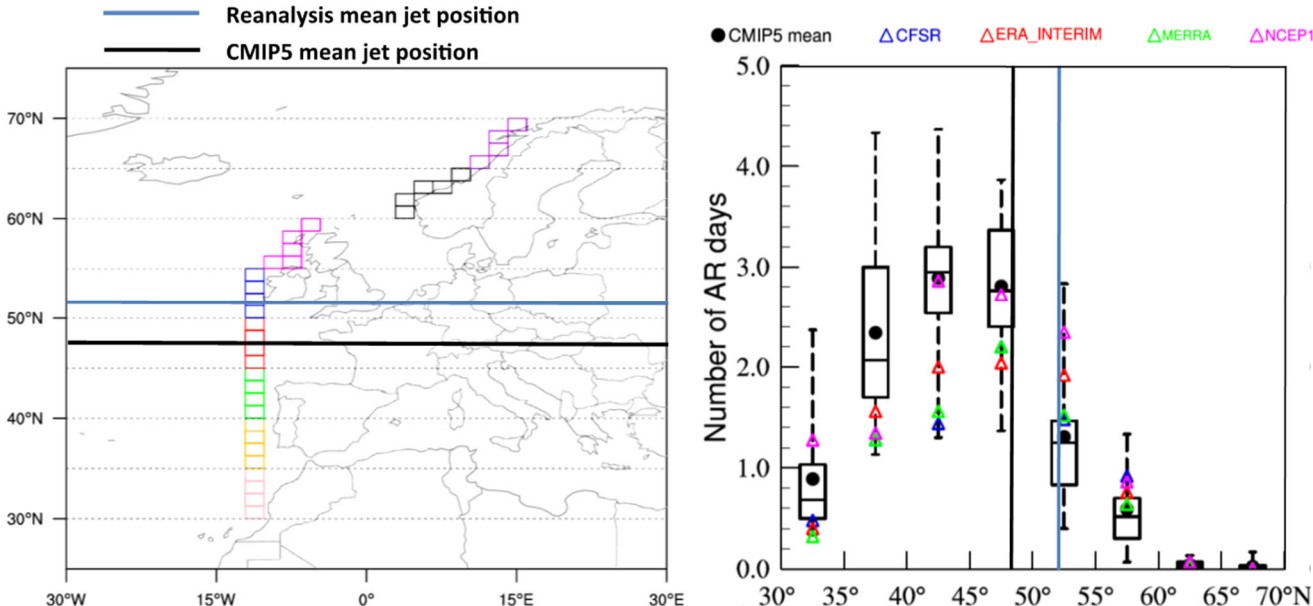

Figure 5.  Atmospheric rivers (ARs) are responsible for over 90% of the moisture transport to the extratropics (Zhu and Newell 1998). They also contribute significantly to heavy precipitation and flooding in many regions worldwide (Ralph et al. 2006). Understanding how ARs may change in a warmer climate is important for managing water resources and flood risk. Associated with Rossby wave breaking, the frequency of ARs and their landfall locations are influenced by the jet stream. Global climate models in the Coupled Model Intercomparison Phase 5 (CMIP5) exhibit an equatorward bias in the simulated jet position. For example, the left panel shows the grid boxes (colored) used to detect CMIP5 model-simulated North Atlantic ARs making landfall in Europe. The black and blue horizontal lines show the CMIP5 and reanalysis mean jet positions, respectively. The CMIP5 models simulate a mean jet stream position that is almost 5° equatorward of that depicted in the reanalysis, probably due to their relatively coarse model resolutions (e.g., Lu et al. 2015). Biases in the jet position have important implications for the simulation of ARs in Europe. As shown in the right panel, CMIP5 models simulated too few (too many) ARs poleward (equatorward) of the observed jet position in the North Atlantic during December-February compared to four global reanalyses (color symbols). Here, the box-and-whisker plots show the CMIP5 multi-model mean (dot), median (horizontal bar), 75% and 25% percentiles (upper and lower boundaries of the box), and the highest and lowest values (whiskers). A challenge for improving the simulation of ARs and their response to warming is the more accurate simulation of the jet stream and the associated Rossby wave dynamics. Enabled by advances in computational resources, increasing model resolution may improve the fidelity of model simulated jet, which may improve projections of changes in extreme precipitation and flooding in a changing climate. [Contact: Ruby Leung.  See from Gao et al. (2016) for more information.]

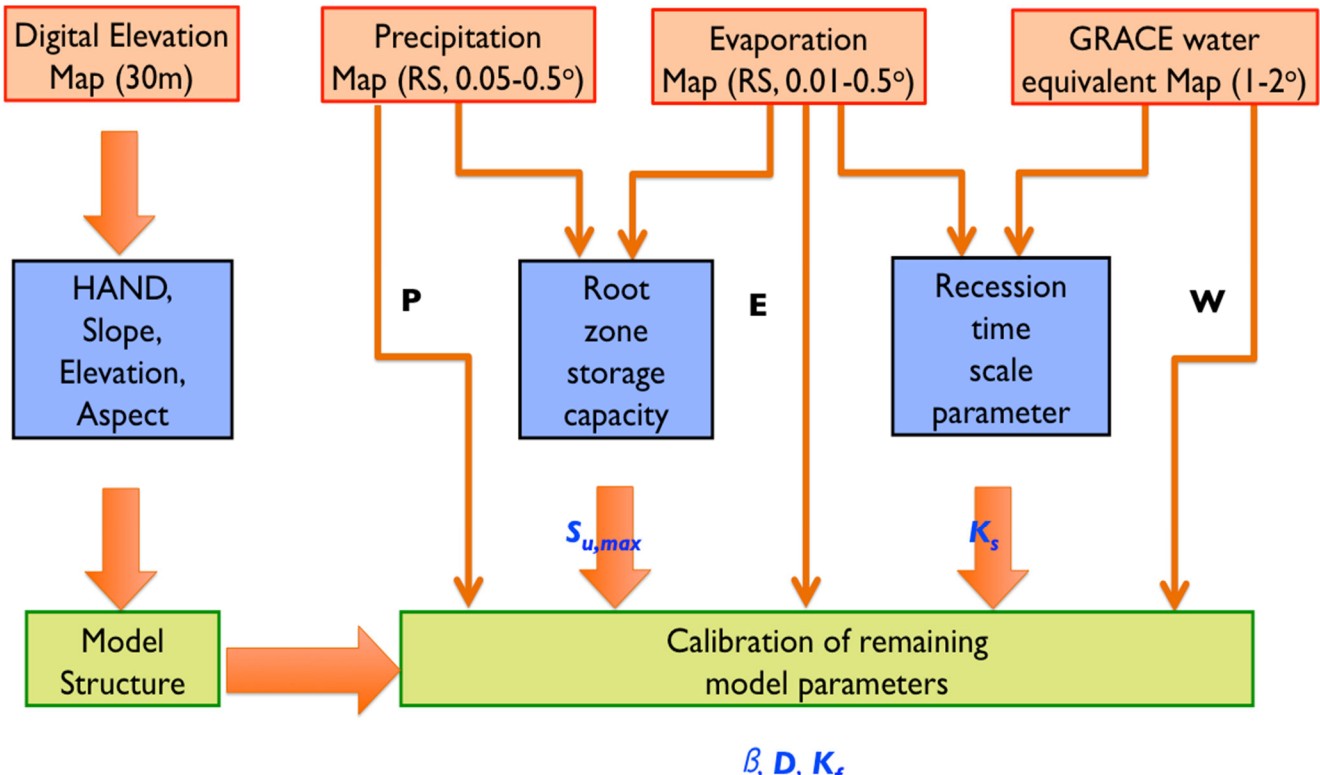

Figure 6. Readily available remote sensing products can be used to constrain hydrological models in a way that allows streamflow prediction in ungauged basins. The above schematic shows the relevant connections to consider during a calibration procedure. HAND refers to the Height Above the Nearest Drainage (which is the hydraulic head), root zone storage capacity is the maximum amount of soil water that can be accessed by the vegetation root systems, and the recession time scale parameter controls the steepness of the recession. P, E, and W represent precipitation, evaporation, and soil water content, with RS indicating a remotely sensed source. $S_{u,max}$ is the root zone storage capacity, $K_s$ is the slow recession time scale, and $\beta$, D, and $K_f$ are the exponent of the threshold function for runoff generation, the splitter between recharge and runoff, and the fast recession time scale, respectively. Note that the root zone storage capacity of ecosystems reflects in part the ability of vegetation to distribute its roots to optimize soil water usage. Through the calibration scheme shown above, we can use historical time series of precipitation and evaporation to derive the effective storage capacity utilized by the ecosystem and then connect it to the ecosystem's survival strategy (Gao et al., 2014). In addition, through such an approach, we can investigate how ecosystems will adjust their storage capacity in response to climatic change and how rainfall-runoff relations will change as a result. [Contact: Hubert Savenije. See Savenije and Hrachowitz (2016, 2017) for more information.]

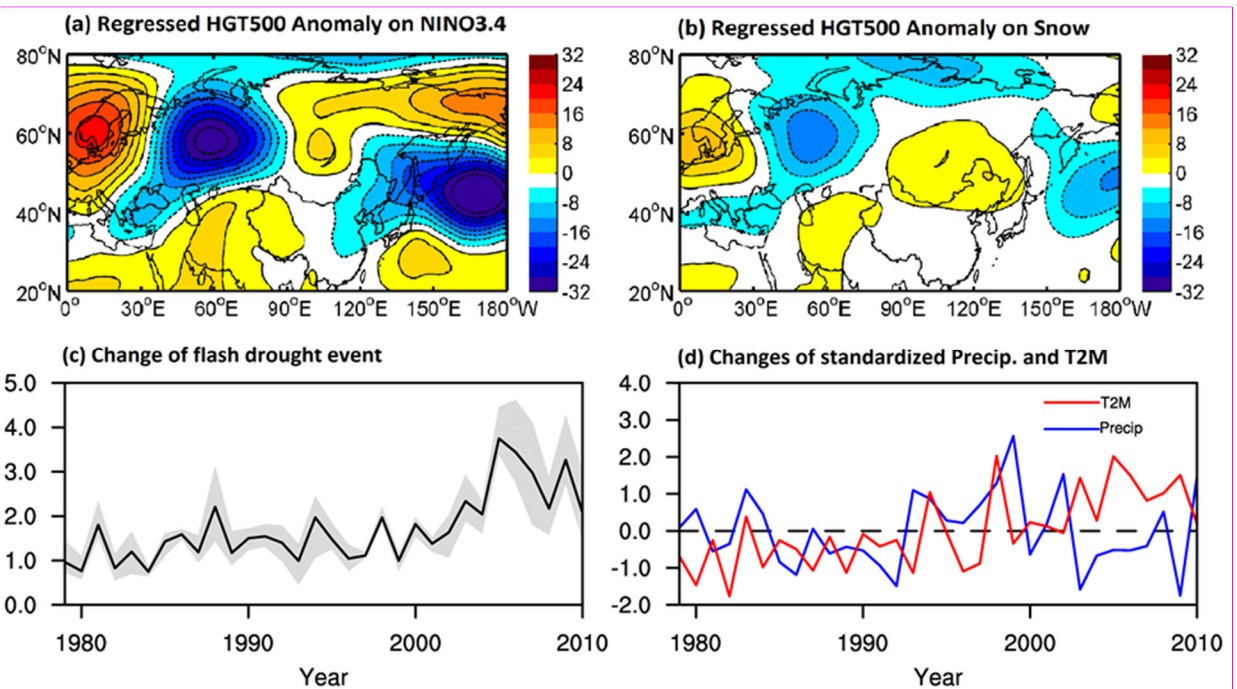

**Figure 7.** Joint analysis of a variety of climate variables provides new insights into the predictability of seasonal drought in China and into recent changes in the character of flash drought there. The top panels
show (a) the slopes (in geopotential meters, or gpm) of the regressions of July-August 500 hPa geopotential height anomaly on detrended (and standardized) July NINO3.4 index and (b) the slopes (also in gpm) of the regressions of this height anomaly on negative (and standardized) March Eurasian snow cover. The two panels demonstrate that both ENSO and Eurasian snow cover are statistically tied to the Eurasia teleconnection (EU) pattern responsible for summer droughts in northern China (modified from
Wang et al., 2017). Note that a seasonal climate forecast model usually shows higher forecast skill during ENSO years; the CFSv2 model, for example, predicted the 2015/16 El Niño and roughly captured the devastating North China drought in the summer of 2015. However, a strong El Niño does not necessarily result in an extreme drought in North China, since such drought also depends on whether the El Niño evolves synergistically with Eurasian spring snow cover reduction to trigger a positive summer Eurasian
teleconnection (EU) pattern (a-b) that favors anomalous northerly air sinking over North China (see Wang et al. 2017 for more information). Regarding changes in the character of flash drought, the two bottom panels show (c) changes in flash drought events (events per year) over southern China and (d) changes in standardized (and thus dimensionless) precipitation and surface air temperature averaged over southern China. The increasing trend in flash drought over southern China suggests that the probability of
concurrent heat extremes, soil moisture deficits, and positive evapotranspiration anomalies there is increasing (see Wang et al., 2016 for more information). [Contact: Xing Yuan]

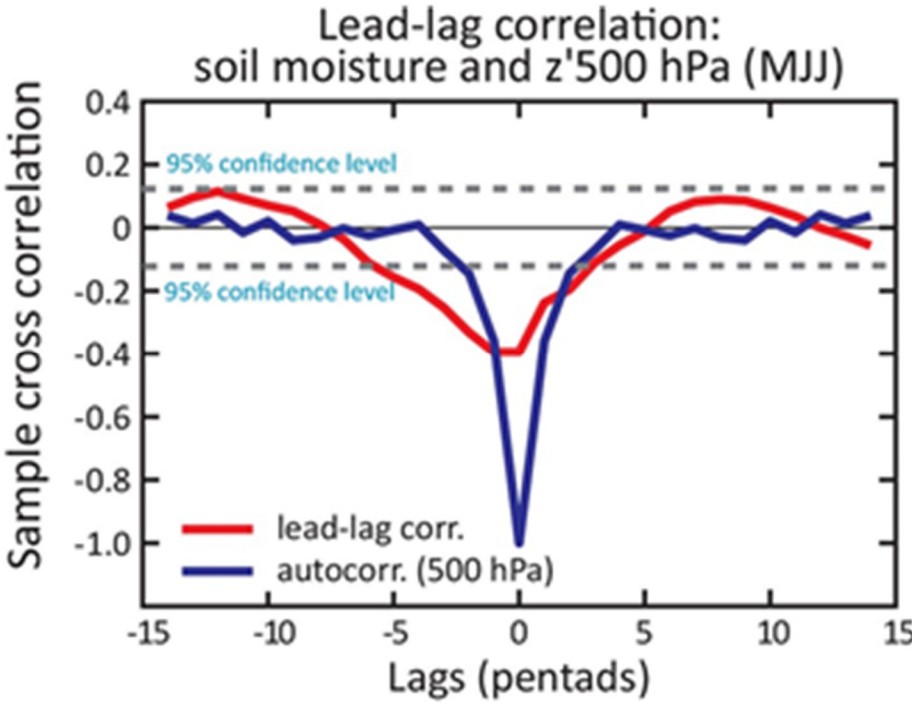

Figure 8.   The possibility that soil moisture anomalies can affect the character of the overlying atmospheric circulation could have profound implications for our understanding of drought evolution and maintenance.  The plot above shows the statistical connection between soil moisture (as derived from offline land analyses) and 500 hPa geopotential height anomalies (as derived from an atmospheric reanalysis).  More specifically, the red curve shows the lead-lag correlation between pentad soil moisture anomalies and the height anomalies during May-July (MJJ) over the south-central United States over the period 1981-2012, whereas the blue line depicts the autocorrelation function (ACF) of the pentad 500 hPa geopotential height anomalies of MJJ for the same region and period.  The ACF values have been multiplied by -1 for easy comparison with the red curve.  The 95% confidence bounds are derived as the standard deviations divided by the square roots of N, where N is the effective number of independent samples.  (The original sample size is n=612, whereas N=139 after accounting for autocorrelation in the time series.)  The fact that the red curve lies below the blue curve (and is significant) for -1 to -6 pentads indicates that positive large-scale mid-tropospheric geopotential height anomalies (which are characteristic of circulation patterns associated with drought) are more correlated with soil moisture deficits 5-30 days earlier than they are with earlier height anomalies, suggesting that the patterns may be influenced more by soil moisture than by the memory of the large-scale atmospheric circulation (either remotely forced by SSTA or through memory provided by the internal atmospheric variability).  This result provides observational evidence of soil moisture feedback on large-scale drought circulation in summer over the south central US (or southern Plains).  [Contact: Rong Fu.  Figure taken from Fernando et al. (2016); see this reference for more information.]

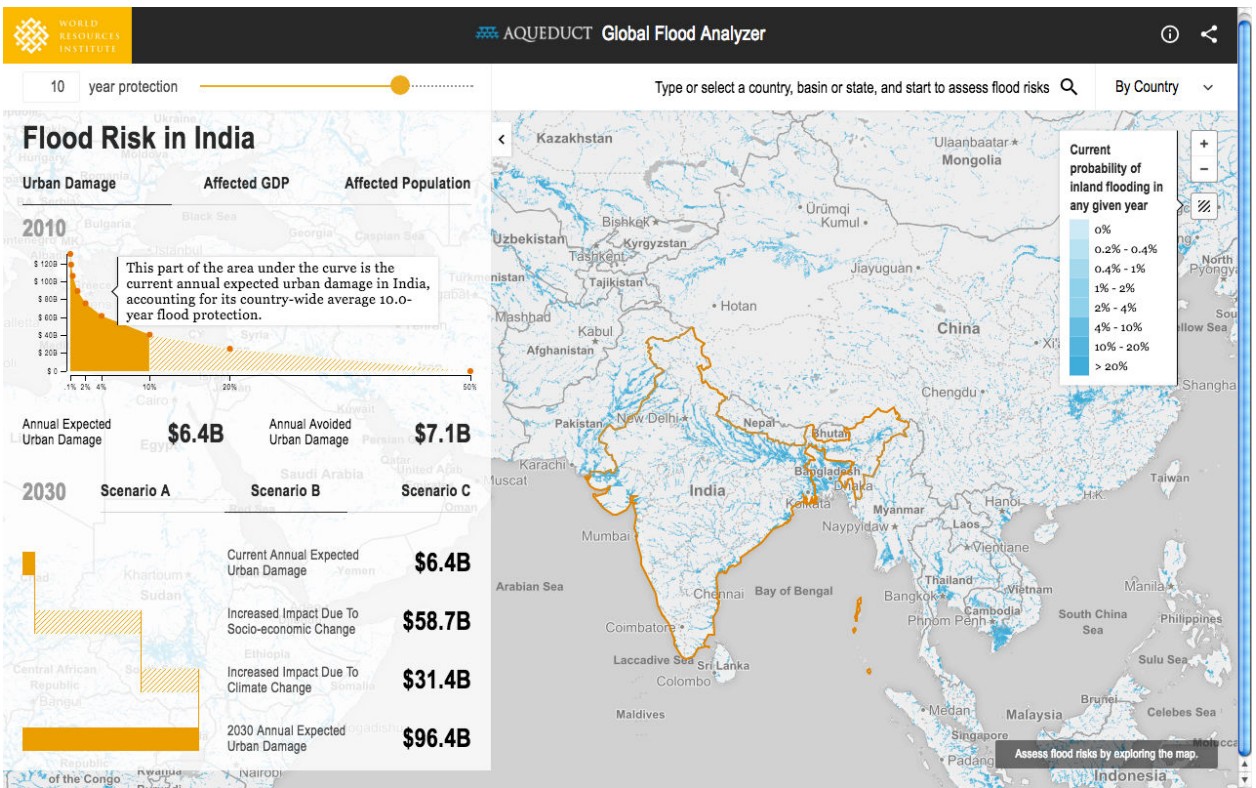

Figure 9. Scientific progress in conjunction with advances in web-based software technologies are providing society with valuable new tools for coping with the physical and economic uncertainties associated with flooding. The above screenshot, for example, is from the Aqueduct Global Flood Analyzer, a web-based interactive platform that estimates river flood risk in terms of urban damage, affected GDP, and affected population at the country, state, and river basin scale across the globe. The Analyzer enables users to estimate current flood risk for a specific geographic unit, taking into account existing local flood protection levels. It also allows users to project future flood risk under climate and socio-economic change and separately attribute change in flood risk to each of these drivers. Finally, for each flood protection level, high-resolution maps of yearly flooding probability are provided. The basis for the Analyzer is the global hydrology and water resources model PCR-GLOBWB (Van Beek et al., 2011). The methodology behind the tool is described extensively in Ward et al. (2013) and Winsemius et al. (2015). Current developments for this tool entail adding the risk of coastal flooding and analyzing the costs and benefits of adaptation measures, including traditional "hard defenses" and nature-based solutions. (Adapted from Bierkens [2015]. Contact: Marc Bierkens)

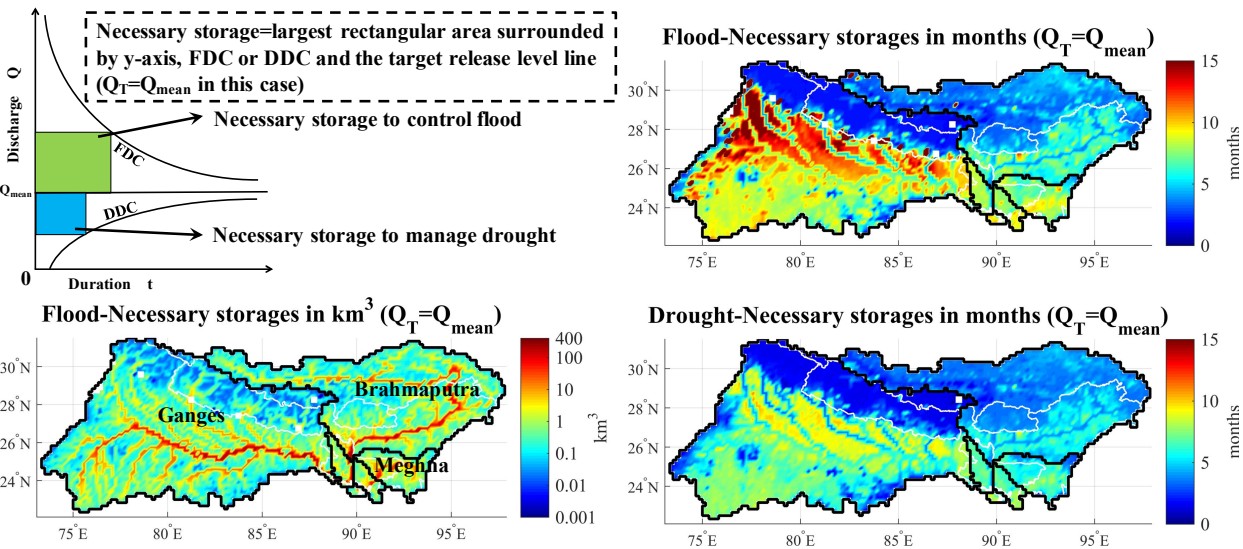

Figure 10: In nature, changes in the storage of water in a hydrological basin can smooth out hydrological variations associated with floods and droughts. The spatial variability in necessary hydrological storage, however, remains relatively unstudied – at the present time there is no global map showing the storage needed to ameliorate floods and droughts, either for the present climate or under climate change. In the panels above, using the Ganges-Brahmaputra-Meghna basin as an example, the needed storage at each grid cell within the basin is calculated with a new method: intensity-duration-frequency curves of flood and drought (flood duration curve and drought duration curve: FDC-DDC, an alternative representation of discharge time series obtained from a calibrated hydrological model called BTOPMC – see Takeuchi and Masood, 2016). For simplicity, the target release ($Q_T$) for smoothing is assumed to be the long term mean discharge ($Q_{mean}$) at each grid cell (Takeuchi and Masood, 2016). The figure shows a typical FDC-DDC curve for a grid cell and an illustration of how to calculate necessary storage (top left), the spatial distribution of storage (in units of km$^3$) needed to smooth floods in the basin (bottom left), and the spatial distribution of storage (in units of months) needed to smooth flood (top right) and drought (bottom right). Note that storages expressed in months, calculated by dividing the necessary storage volume by the local $Q_{mean}$ for 1979-2003, provide a unique perspective on storage requirements. The geographical distribution of necessary storage reflects hydrological heterogeneity associated with meteorological inputs, topography, geology, soil, vegetation, landuse, and so on. Quantifying the relationships between spatially distributed necessary storages and the geographical distribution of hydroclimatological, geological and land cover conditions can lead to improved hydrological analysis and produce useful information for water resources managers. (Contact: Muhammad Masood )

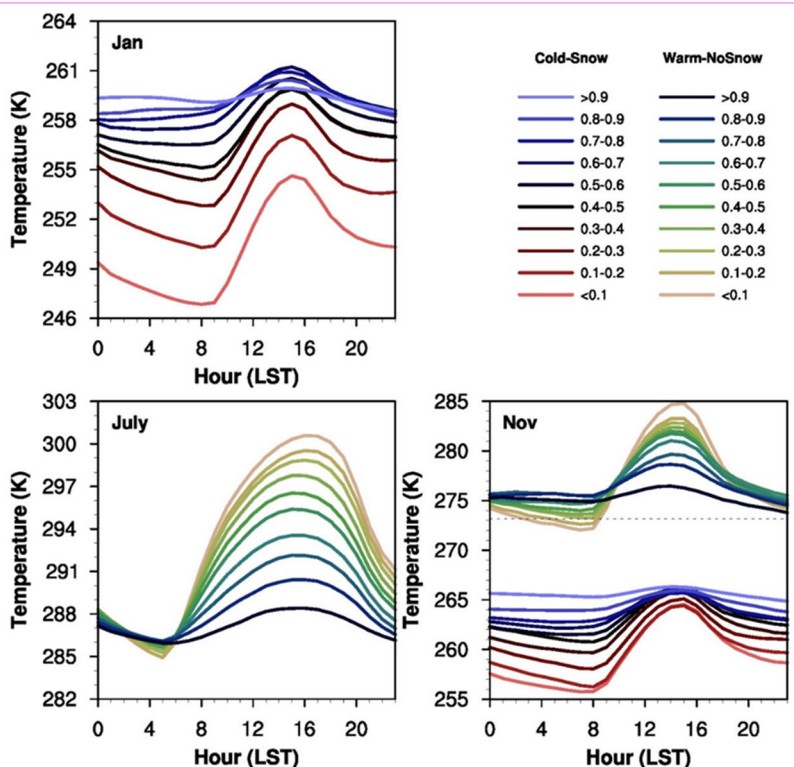

Figure 11. Land surface hydrological processes and atmospheric (boundary layer) processes do not proceed in isolation from each other; land states and boundary layer states evolve together, as a joint system. The nature of this coupled system was recently elucidated through a careful analysis of a wealth of land surface and boundary layer data collected by trained observers in the Canadian Prairies. These observers recorded hourly, since 1953, the fraction of the sky covered by opaque reflective cloud, providing daily shortwave and long-wave cloud forcing (SWCF and LWCF) on climate timescales when calibrated against baseline surface radiation measurements (Betts et al. 2015). The panels above express some of the important relationships inherent in these data in the form of average diurnal temperature cycles for January (top left), July (bottom left), and the fall transition month of November (bottom right). For each month, days are binned by daily mean opaque cloud fraction in tenths, with a different color scheme for cold days with mean temperature <0ºC and snow cover, and days >0ºC and no snow cover. In July, the diurnal cycle of temperature and relative humidity is dominated by SWCF on both daily and monthly timescales, and temperatures rise under clear skies. In contrast, in January the temperatures are lower under clear skies as LWCF dominates (Betts et al. 2014a, 2015). It is in fact the presence or absence of reflective snow cover that determines the impact of clouds on surface temperature – in November, the snow-free days are more than 10K warmer than the snow-covered days, and the former shows the July type of behavior whereas the latter shows the January type of behavior. [Contact: Alan Betts. Adapted from Betts and Tawfik (2016).]

| | Bayesian | Variational | OPTIMISTS |
|---|---|---|---|
| **Resulting state-variable estimate** | Gaussian (KF, EnKF), Non-Gaussian (PF) | Deterministic (unless adjoint model is used) | Non-Gaussian |
| **Solution quality criteria** | High likelihood given observations | Minimum cost value (error, consistency) | Minimum error, maximum consistency with history |
| **Assimilation time step** | Sequential | Sequential (1D-3D) or entire window (4D) | Flexible |
| **Search method** | Iterative Bayesian belief propagation | Convex optimization | Coupled belief propagation/multi-objective optimization |
| **Model dynamics** | Linear (KF), non-linear (EnKF, PF) | Linearized to obtain convex solution space | Non-linear (non-convex solution space) |

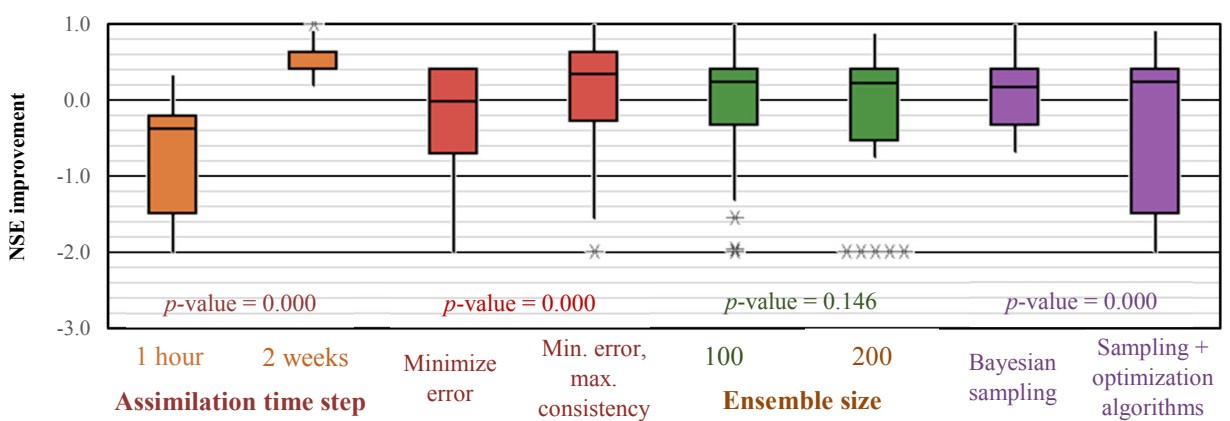

Figure 12. The success of hydrological prediction depends largely on the accuracy of the initialization of the forecast model. Advanced mathematical tools (i.e., data assimilation algorithms) are now available to transform a given set of observations into the best forecast initialization possible. The table above outlines the features of three data assimilation approaches: standard Bayesian data assimilation algorithms (KF stands for Kalman Filter, EnKF stands for Ensemble Kalman Filter, and PF stands for Particle Filter), variational methods, and a new technique – OPTIMISTS – that combines the advantageous characteristics of the first two. Some of the features selected for OPTIMISTS, such as non-Gaussian probabilistic estimation and support for non-linear model dynamics, are considered advantageous in the literature (van Leeuwen, 2015); flexible configurations are available for other features (e.g., the choice of optimization objectives or the analysis time step) for which no consensus has formed. In the bottom panel, different configurations of OPTIMISTS (indicated along x-axis) are compared in terms of their success in improving streamflow forecasts. The experiments were conducted with the Distributed Hydrology Soil Vegetation model (DHSVM) on a test case with 1,472 cells and over 30,000 state variables; the ordinate shows the change, relative to a control that uses no data assimilation, in the Nash-Sutcliffe Efficiency (NSE) coefficient (positive values indicating forecast skill improvement). Asterisks on the boxplots indicate outliers. Three configurations of OPTIMISTS provide statistically significant advantages (demonstrated by the indicated p-values from the ANalysis Of VAriance): (i) setting the analysis time step equal to the entire two-week assimilation period; (ii) maximizing the consistency of the states with the background (and not only minimizing the error); and (iii) using only Bayesian sampling to generate new members/particles. Studies like this are critical for maximizing the effectiveness of the techniques used to initialize forecast models; this particular study positions OPTIMISTS as a capable and flexible framework. [Contact: Xu Liang.]

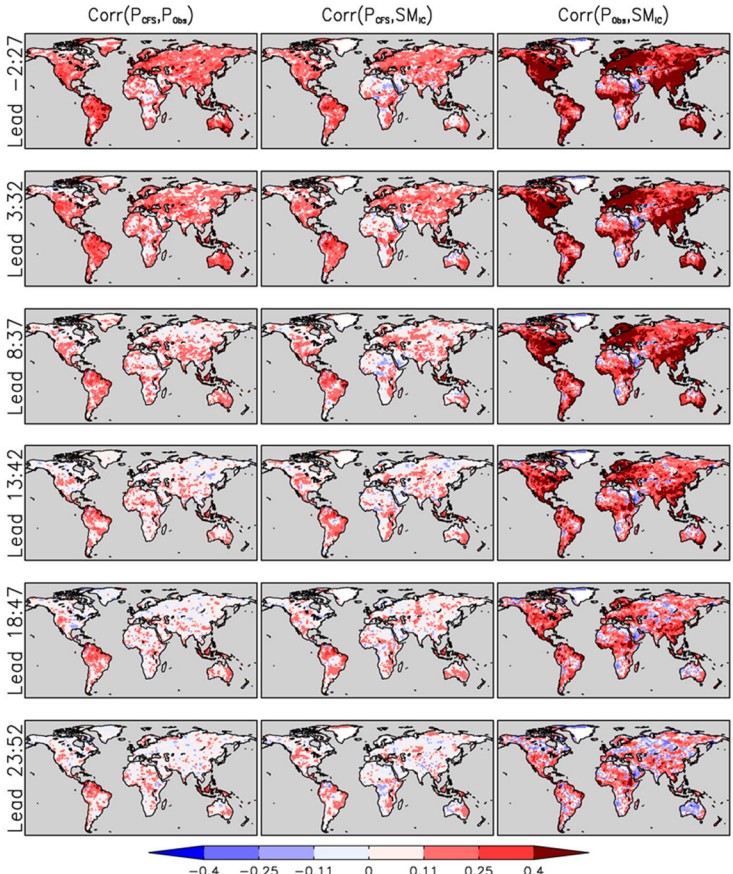

Figure 13. If, in the real world, land surface variations (e.g., in soil moisture) are able to affect the overlying atmosphere, and if an atmospheric model does not capture adequately this land-atmosphere

feedback, the performance of the model will suffer. A forecast model that lacks this feedback likely cannot translate the information contained in soil moisture states into improved forecasts of air temperature and precipitation. With this as motivation, the panels above provide an evaluation of land-atmosphere feedback in the US operational forecast model (CFSv2). The three columns show from left to right the pair-wise correlations (i) between monthly CFSv2 reforecast precipitation ($P_{CFS}$) and observed

precipitation ($P_{Obs}$), (ii) between $P_{CFS}$ and reforecast initial soil moisture in layer 2 (10-40cm depth; $SM_{IC}$), and (iii) between $P_{Obs}$ and $SM_{IC}$, all for forecasts validating during JJA. The rows show the different leads (in days) considered. Dark colors (beyond ±0.11) are significant at the 95% confidence level. The fact that observed precipitation rates are more closely related to antecedent soil moisture than are model simulated rates suggests that the US operational forecast model underestimates land-atmosphere coupling.

An improvement in the system's simulation of coupled land-atmosphere processes could improve the accuracy of the forecasts produced. . [Contact: Paul Dirmeyer. Figure taken from Dirmeyer (2013); see this reference for further information]