# Peer review of "Hydroclimatic Variability and Predictability: A Survey of Recent Research"

_Hydrology and Earth System Sciences, 2017_

## Referee Comment (RC1) · B. van den Hurk (Referee) · 29 Mar 2017

This is a somewhat unusual but useful format for a scientific paper, as explained by the authors: not intented as an extensive review of the dynamic field of research in hydrometeorological predictions, but an instructive topical description of some relevant aspects of the scientific field, illustrated by studies which were presented at Eric Woods honor symposium.

The body of the paper is well written an well structured, but the degree to which the selected studies are self-explanatory varies somewhat at could be improved (see detailed comments below). Some figures showed very interesting results (8, 10, 11).

Detailed comments

- p2, l26: also the fact that many hydrological catchments cross national boundaries, and that cascades in impacts take place that are not limited to individual catchments is a reason to address this topic using large scale models

- p3, l28: you might mention that also developments in the observational records and techniques have contributed to this progress

- p4, l15: the reason why this modelling at hyperresolution would be beneficial could be mentioned here, it is not obvious

- p5, l16: please replace "predictions" by "projections"

- p5, l22 and Figure 2: the degree to which "local" phenomena are explained by "nearby" drivers is also a matter of definition of scales. If one is interested in precipitation variability at 50x50km resolution then it is obvious that nearby drivers have a large impact. Aggregation to the spatial scale of entire continents, however, makes also inland territories sensitive to ENSO-like drivers

- p8, l9: typo in "Berghuijs"

- p12, l28: which "Dirmeyer (2013)" is referred to here? There are 3 papers Dirmeyer 2013

- Summary: a reference to a website where the original presentations can be downloaded would be a valuable addition to this paper

Comments per figure

- Figure 3: the textual explanation is fairly thin: it is unclear what kind of model upgrade was applied, and whether the bias correction of VICET or the bias correction of the NARCCAP was the dominant factor in explaining the differences shown

- Figure 6: many of the terms mentioned in the figure are not explained. Also it is unclear what is meant with "living agents"

- Figure 7: what are the units of the contours shown?

- Figure 13: the units of the lead time shown on the left is unclear.

[Figure]

---

## Referee Comment (RC2) · Anonymous Referee #2 · 14 Apr 2017

The paper "Hydroclimatic Variability and Predictability: A Survey of Recent Research" provides and overview of recent research in large-scale hydroclimatic variability. This includes general variability, droughts, floods, land-atmosphere interactions and hydro-climatic prediction. For each of these subcategories a summary of recent research and examples from a recent symposium are presented. The paper has a very clear and instructive outline that makes the paper interesting and understandable. The authors also do a good job of providing an overview of recent research without it becoming overwhelming. However, there are two key weaknesses that need to be addressed before the paper is published.

First, the examples from the symposium are poorly constructed and the lack of detail makes them seem unimportant and irrelevant to the overall paper. For example, section 2.1.2 presents 6 different and seeming unrelated figures in only 13 lines of text. There

needs to be more discussion in the paper about each of the figures and how it relates to the section theme. It would be helpful if there was a short summary at the end of each example section that discussed the connection between figures. It also seems that the authors tried to compensate for the lack of discussion in the text by making the figure captions overly detailed. This is cumbersome to read and makes the figures disconnected from the paper. For example, the caption for Figure 10 consists of 262 words while the example section 2.3.2 which references the figure only contains 87 words. The paper would be greatly improved if the authors revised all the example sections to include a more coherent structure that offers more detail and connection to the section. This may require cutting the number of examples for some sections, but fewer well discussed figures that integrate with the rest of the paper would be more useful and interesting than simply listing numerous examples.

The second aspect that needs to be improved is the final summary (section 3). The current summary serves little purpose other than complying with the normal writing convention that dictates papers end with a summary. Given that the motivation of this paper was a recent symposium, it seems that the summary should tie all the presented examples together to illustrate the overall thesis of the paper, which seems to be the last sentence in the abstract. While the literature review supports this thesis, the examples presented do not. Also, given that the authors are leading researchers in the subject, the summary would be greatly improved by discussing challenges and future directions for the field.

Overall, this paper is interesting and meaningful and should be published, however, given the two major weaknesses discussed, it still needs revisions that ranges between minor and major.

---

## Referee Comment (RC3) · Anonymous Referee #3 · 17 Apr 2017

A recent symposium honoring Eric Wood and featuring many luminaries provides a moment to pause and consider the state of research on hydroclimatic variability and predictability. This paper exploits that opportunity reasonably well, though not as fully as it might. With a little extra work on the example figures–as explained below–this could be a great introduction/guidance to newer researchers in the field.

The overall structure of this survey seems carefully thought out and is well executed. The writing in the main text is unusually fluid and accessible, and the disparate blurbs from the various authors are woven in (at least) as well as could be expected. I would suggest another iteration on the figures, with the captions being more like complete and self-contained abstracts that happen to be accompanied by graphics. Each caption could start with an introductory sentence or two motivating the work, then describe the work, more than superficially. Next, report a main finding, with reference to the figure.

[Figure]

(All the symbols, axes, units in each figure, of course, should be defined/explained.) Finally, one might hope that each of the contributing authors would close his/her abstract with a sentence or two of wider perspective of the "challenges and directions" sort that this example helps to motivate. Ideally, the paper's overall summary would somehow distill these perspectives into some general conclusions.

---

## Author Comment (AC1) · 25 May 2017

» This is a somewhat unusual but useful format for a scientific paper, as explained by the authors: not intented as an extensive review of the dynamic field of research in hydrometeorological predictions, but an instructive topical description of some relevant aspects of the scientific field, illustrated by studies which were presented at EricWoods honor symposium. The body of the paper is well written an well structured, but the degree to which the selected studies are self-explanatory varies somewhat at could be improved (see detailed comments below). Some figures showed very interesting results (8, 10, 11).

Thank you. Specific responses are provided below.

»Detailed comments »p2, l26: also the fact that many hydrological catchments cross

[Figure]

national boundaries, and that cascades in impacts take place that are not limited to individual catchments is a reason to address this topic using large scale models

We address this with the following addition: "Many important hydrological problems must be addressed at the large basin scale, a scale that transcends political boundaries and is not amenable to techniques designed for traditional small-scale catchments."

» p3, l28: you might mention that also developments in the observational records and techniques have contributed to this progress

The reviewer is right. We have amended the text as follows (new text underlined): "...the continually growing availability of powerful computational tools (along with more extensive observational records and improved analysis techniques) for examining this variability..."

» p4, l15: the reason why this modelling at hyperresolution would be beneficial could be mentioned here, it is not obvious

We amended the sentence to read (new text underlined): "Wood et al. (2011) emphasize the importance to society of developing hyper-resolution ($\leq$1 km resolution) land surface modeling systems at continental to global scales; such resolutions would allow an improved representation of the impacts of spatial heterogeneity in surface properties on large-scale hydrological and atmospheric dynamics."

» p5, l16: please replace "predictions" by "projections"

Done.

» p5, l22 and Figure 2: the degree to which "local" phenomena are explained by "nearby" drivers is also a matter of definition of scales. If one is interested in precipitation variability at 50x50km resolution then it is obvious that nearby drivers have a large impact. Aggregation to the spatial scale of entire continents, however, makes also inland territories sensitive to ENSO-like drivers

True. We have changed the main text to emphasize that we are not speaking of Australia as a whole (new text underlined): "In another study, variability of rainfall over parts of Queensland, Australia, is found to be potentially controlled more by nearby sea surface temperatures (SSTs) than by distant climate phenomena such as El Niño (Figure 2)." We have also amended the caption to read (new text underlined): "In many places the high local SSTs (within a few hundred km of the coast) accounted for more of the precipitation than the prevailing La Niña conditions did at the spatial scales considered here."

» p8, l9: typo in "Berghuijs"

Corrected.

» p12, l28: which "Dirmeyer (2013)" is referred to here? There are 3 papers Dirmeyer 2013

Actually, there are three Dirmeyer et al (2013) papers, but only one Dirmeyer (2013) paper. We think the figure is referred to correctly as is.

» Summary: a reference to a website where the original presentations can be downloaded would be a valuable addition to this paper

We agree, but we have no control over how long these presentations will be maintained on the workshop website – this would be subject to the whims of the folks at Princeton. While we were told that there are no immediate plans to remove the presentations, we think that providing what may soon be a dead link is inappropriate. If the editor wants us to include the link, we can. For now, we leave it out. In any case, most of the figures point to papers that contain additional information, and all of the papers point to a contact that can be reached for further information.

» Comments per figure

» Figure 3: the textual explanation is fairly thin: it is unclear what kind of model upgrade was applied, and whether the bias correction of VICET or the bias correction of the

NARCCAP was the dominant factor in explaining the differences shown

The caption has been overhauled. The VICET model overwrites the VIC-estimated ET components using bias-corrected values, which has the effect of improving the estimation of other hydrological variables as well, as now explicitly stated in the updated figure caption. The differences in hydrological behavior between the two simulations have nothing to do with the bias correction of NARCCAP forcing, as the two simulations utilize the same meteorological forcing. This has been clarified in the updated figure caption.

» Figure 6: many of the terms mentioned in the figure are not explained. Also it is unclear what is meant with "living agents"

The caption for figure 6 has been expanded to explain all terms. The reference to "living agents" was unnecessary and has been removed.

» Figure 7: what are the units of the contours shown?

Units are now provided in the caption.

» Figure 13: the units of the lead time shown on the left is unclear. The units are now provided in the caption.

---

## Author Comment (AC2) · 25 May 2017

» The paper "Hydroclimatic Variability and Predictability: A Survey of Recent Research" provides and overview of recent research in large-scale hydroclimatic variability. This includes general variability, droughts, floods, land-atmosphere interactions and hydro-climatic prediction. For each of these subcategories a summary of recent research and examples from a recent symposium are presented. The paper has a very clear and instructive outline that makes the paper interesting and understandable. The authors also do a good job of providing an overview of recent research without it becoming overwhelming. However, there are two key weaknesses that need to be addressed before the paper is published.

Thank you. See responses below.

» First, the examples from the symposium are poorly constructed and the lack of detail makes them seem unimportant and irrelevant to the overall paper.

All of the captions have been enhanced substantially; the figure/caption combinations are now more self-contained and complete. (See response to Reviewer 3.)

» For example, section 2.1.2 presents 6 different and seeming unrelated figures in only 13 lines of text. There needs to be more discussion in the paper about each of the figures and how it relates to the section theme. It would be helpful if there was a short summary at the end of each example section that discussed the connection between figures.

We agree that the approach used in the original manuscript was deficient, the text describing the figures being written in a vaguely "stream-of-consciousness" way. In the revised manuscript, the examples are presented in bullet form, surrounded by brief introductory and closing comments that connect the figures, at least to the fullest extent possible. This, we feel, clarifies the role of the figures in this paper – as representative examples of research, not as a carefully selected set of related results that, when considered together, illustrate some larger point. By bulletizing the figures and their contents, and by treating the bulleted figure/caption combinations as self-contained entities, we give the reader the freedom to explore the particular figures of interest to him or her.

» It also seems that the authors tried to compensate for the lack of discussion in the text by making the figure captions overly detailed. This is cumbersome to read and makes the figures disconnected from the paper. For example, the caption for Figure 10 consists of 262 words while the example section 2.3.2 which references the figure only contains 87 words. The paper would be greatly improved if the authors revised all the example sections to include a more coherent structure that offers more detail and connection to the section. This may require cutting the number of examples for some sections, but fewer well discussed figures that integrate with the rest of the paper would

be more useful and interesting than simply listing numerous examples.

Again, we agree that the original presentation of the figures was deficient. The reviewer provides here a viable approach for improvement. We note, however, that Reviewer 3 suggests a distinct strategy, in the opposite direction: the reformulation of the figures/captions as "complete and self-contained abstracts that happen to be accompanied by graphics." This latter strategy in fact agrees with our original vision of the paper. We never meant, as suggested above, to compensate for a lack of discussion in the text by making the captions overly detailed; we meant all along to make each figure self-contained and to keep the associated discussion in the main text to a bare minimum. This is now done more successfully through an overhauled set of captions and through the aforementioned reworking of the "examples from the symposium" sections. In a sense, we consider the figures in the paper to be equivalent to "sidebars" in a report, sidebars that stand alone and need not tie together to make a broader story. Thus, while we have considered this reviewer's suggestion seriously, we have instead followed Reviewer 3's suggestion. Removing figures in any case would mean removing authors from the author list, something we are loath to do.

» The second aspect that needs to be improved is the final summary (section 3). The current summary serves little purpose other than complying with the normal writing convention that dictates papers end with a summary. Given that the motivation of this paper was a recent symposium, it seems that the summary should tie all the presented examples together to illustrate the overall thesis of the paper, which seems to be the last sentence in the abstract. While the literature review supports this thesis, the examples presented do not. Also, given that the authors are leading researchers in the subject, the summary would be greatly improved by discussing challenges and future directions for the field.

The reviewer is absolutely correct; the summary section in the original manuscript was lacking. We have expanded this section in a way that we hope places the research discussed in the broader context of critical challenges and new opportunities in the

field.

» Overall, this paper is interesting and meaningful and should be published, however, given the two major weaknesses discussed, it still needs revisions that ranges between minor and major.

---

## Author Comment (AC3) · 25 May 2017

» A recent symposium honoring Eric Wood and featuring many luminaries provides a moment to pause and consider the state of research on hydroclimatic variability and predictability. This paper exploits that opportunity reasonably well, though not as fully as it might. With a little extra work on the example figures–as explained below–this could be a great introduction/guidance to newer researchers in the field.

Thank you. See our responses below.

» The overall structure of this survey seems carefully thought out and is well executed. The writing in the main text is unusually fluid and accessible, and the disparate blurbs from the various authors are woven in (at least) as well as could be expected.

[Figure]

Thank you. (No response necessary.)

» I would suggest another iteration on the figures, with the captions being more like complete and self-contained abstracts that happen to be accompanied by graphics. Each caption could start with an introductory sentence or two motivating the work, then describe the work, more than superficially. Next, report a main finding, with reference to the figure. (All the symbols, axes, units in each figure, of course, should be defined/explained.) Finally, one might hope that each of the contributing authors would close his/her abstract with a sentence or two of wider perspective of the "challenges and directions" sort that this example helps to motivate.

We have followed the reviewer's advice and have overhauled all of the figure captions. Motivations are included, results are discussed, and results are put into broader hydrological perspective. Units, symbols, etc., are explained. Through this overhaul, we feel that the information and insights contained in the different figures are now much more accessible to the reader than they were in the original manuscript.

» Ideally, the paper's overall summary would somehow distill these perspectives into some general conclusions.

The summary section has been expanded considerably; it now places the research discussed in the broader context of hydrological challenges and opportunities.